# Amphotericin B promotes respiratory viral entry by enhancing late endosomal maturation and fusion via glucocerebrosidase-mediated ceramide remodeling

Respiratory viral infections, such as influenza and COVID-19, pose significant global health challenges. For patients with invasive pulmonary aspergillosis, a subsequent viral infection can lead to markedly worse clinical outcomes. Although amphotericin B (AmB) remains a cornerstone antifungal therapy, our investigation demonstrates that it paradoxically enhances the entry of influenza A virus and SARS-CoV-2. Mechanistically, AmB directly binds to and activates glucocerebrosidase, leading to ceramide accumulation and RAB7 upregulation in the late endosomes, thereby enhancing late endosomal maturation and fusion with viruses. In animal models, AmB treatment enhances viral infection in both influenza A virus–infected mice and SARS-CoV-2–challenged hamsters, resulting in accelerated weight loss, higher viral loads, and aggravated tissue damage. Consistently, in our propensity score-matched cohort of patients with culture-confirmed invasive pulmonary aspergillosis (2016–2025, n = 1,072), systemic use of AmB is associated with a significantly higher incidence of subsequent viral infection compared to other antifungals (21.55% vs. 7.76%, P = 0.003), which is further supported by multivariable analysis confirming AmB as an independent risk factor (adjusted OR = 3.45, 95% CI 2.20–5.41, P = 7.174 × 10⁻⁸). In summary, our findings provide crucial clinical evidence to guide antifungal therapy and reveal glucocerebrosidase as a potential target for developing novel antiviral strategies.

Respiratory viral infections pose a critical global health challenge. Global influenza epidemics account for 290,000–650,000 deaths annually[1], while COVID-19 has caused more than 7 million fatalities as of December 2025[2]. Moreover, invasive fungal coinfections have been linked to a synergistic increase in mortality, further exacerbating the overall disease burden. Strikingly, 15–20% of ICU-admitted patients with severe influenza or COVID-19 develop influenza-associated pulmonary aspergillosis (IAPA) or COVID-19-associated pulmonary aspergillosis (CAPA)[3–7], with case-fatality rates doubling to 50% compared to viral pneumonia alone[7]. COVID-19-associated mucormycosis, though less prevalent, exhibits a 37.4% case-fatality rate in high-risk cohorts[8]. Notably, although the canonical sequence described in the literature is viral infection followed by secondary fungal invasion, recent clinical series have documented the reverse sequence−patients

✉ e-mail: luo_min@fudan.edu.cn; pgwang@ccmu.edu.cn; wzai_163pass@163.com; caobin_ben@163.com

with established invasive pulmonary aspergillosis (IPA) who subsequently acquire SARS-CoV-2 exhibit markedly worse outcomes[9]. While azoles remain first-line therapy for IPA, liposomal amphotericin B (AmB) is guideline-recommended as primary therapy for mucormycosis[10] and azole-resistant aspergillosis[11].

AmB, a polyene macrolide antifungal agent, exerts its fungicidal activity by selectively binding to ergosterol in fungal membranes, forming transmembrane pores that disrupt ion homeostasis[12]. In addition, AmB also interacts with cholesterol in mammalian cell membranes—a property implicated in its nephrotoxicity[13]. Intriguingly, this cholesterol-binding capacity enables AmB to inhibit cholesterol-dependent caveolin-mediated endocytosis, thereby suppressing viral entry of enterovirus 71, Japanese encephalitis virus, and even human immunodeficiency virus type 1 by blocking virus–cell fusion after CD4 binding[14–17].

Paradoxically, emerging evidence demonstrates that AmB facilitates the entry of influenza viruses and SARS-CoV-2 by counteracting interferon-induced transmembrane protein 3 (IFITM3)-mediated restriction[18–20]. Another study showed that nystatin—a structurally similar polyene macrolide targeting ergosterol—contrasts with AmB by potently inhibiting SARS-CoV-2 replication[21]. These contradictory outcomes highlight unresolved mechanistic divergences within this pharmacological class. Furthermore, current evidence remains limited to preclinical models, and whether conventional antifungals could modulate viral pathogenesis to potentially cause therapeutic cross-interference in coinfected patients remains unclear. Hence, further studies are needed to elucidate the virological effects of AmB, aiming to optimize its clinical application.

Here, we identify a previously unrecognized pathway through which AmB facilitates virus entry. AmB directly binds to and activates glucocerebrosidase (GCase), triggering aberrant ceramide accumulation and subsequent remodeling of late endosomal membrane properties to facilitate viral entry. This mechanism is conserved across multiple viral families, including coronaviruses and influenza viruses. Furthermore, AmB exacerbated viral pathogenesis in animal models, and in a retrospective cohort of patients with *Aspergillus* infection, AmB-based therapy was associated with an increased risk of subsequent respiratory viral infection compared with non-AmB regimens. Collectively, our findings establish that AmB is involved in viral pathogenesis and may provide guidance for the drug treatment of fungal infections.

## Results

### AmB exacerbates infection and injury in animal models of influenza and COVID-19

To evaluate the in vivo impact of AmB on respiratory viral infections, we established two animal models: mice challenged with influenza A virus (IAV) and golden hamsters infected with SARS-CoV-2 (Fig. 1A, G). The application of AmB at 1 mg/kg or lower did not cause obvious toxicity in mice and hamsters (Figs. S1A–L, S2A–L). Based on these results, a dose of 1 mg/kg AmB was used for the subsequent experiments.

In the infection assays, AmB was administered 1 h prior to viral challenge and was repeated at 24 and 72 h post-infection (hpi) (Fig. 1A, G). In IAV-infected mice, AmB treatment markedly exacerbated disease progression compared to controls, as evidenced by accelerated weight loss (Fig. 1B) and a significant increase in lung infectious virus titers at both 2 and 5 days post-infection (Fig. 1C). Concurrently, a subtle elevation in serum creatinine levels was observed, suggesting the development of subclinical renal dysfunction (Fig. 1D). Histopathological analysis demonstrated severe pulmonary pathology characterized by extensive necrosis, alveolar hemorrhage, and pronounced leukocytic infiltration (Fig. 1E, F).

Similarly, in the SARS-CoV-2-infected hamster model, AmB-treated animals experienced rapid weight loss (Fig. 1H), increased

viral replication, as indicated by a greater number of nucleocapsid protein-positive cells in lung sections (Fig. 1I, J), and aggravated lung injury marked by diffuse alveolar edema and inflammatory consolidation (Fig. 1K, L). Furthermore, transient increases in urea levels were detected, indicating systemic multi-organ stress responses (Fig. 1M).

### AmB treatment is associated with an increased risk of secondary viral infection

To extend our preclinical findings to a clinical context, we conducted a retrospective cohort study from 2016 to 2025. We included adult inpatients at China-Japan Friendship Hospital with culture-confirmed pulmonary aspergillosis who received systemic antifungal therapy ($n = 1072$) (Fig. S3). To account for potential annual variations in prescribing practices, we present the distribution of cases and AmB use across the study years (Table S1). We first determined the frequency of our primary outcome: a total of 112 patients (10.4%) developed a laboratory-confirmed respiratory viral infection after initiating antifungal therapy, confirming that this temporal sequence is not uncommon in clinical practice.

As detailed in Table S2, patients receiving AmB exhibited more severe baseline illness, indicating substantial confounding by indication. To mitigate this bias, we performed 1:1 propensity score matching (PSM) on key demographic, clinical, and treatment variables. In the matched cohort ($n = 116$ per group), the incidence of post-antifungal viral infection was significantly higher in the AmB group compared to the Non-AmB group (21.55% vs. 7.76%, $P = 0.003$) (Table S3). This association remained robust in a multivariable logistic regression model adjusted for major confounders, including age, sex, comorbidities, and immunosuppressive medication use (adjusted OR = 3.45, 95% CI 2.20–5.41, $P = 7.174 \times 10^{-8}$) (Table S4). A sensitivity analysis based on another PSM cohort ($n = 37$ per group) revealed a consistent trend toward higher risk of secondary viral infection with AmB compared to caspofungin, another second-line anti-*Aspergillus* agent, albeit not statistically significant (13.51% vs. 5.41%, $P = 0.427$). Furthermore, other systemic antifungal agents did not enhance viral entry in vitro, highlighting the distinct profile of AmB (Fig. S4A–D).

### AmB enhances IAV entry without affecting viral attachment or endocytosis

Since AmB could exacerbate respiratory viral infections in vivo, we further conducted in vitro studies to elucidate the underlying mechanisms. We pretreated A549 cells with a non-toxic AmB concentration (Fig. S5A) prior to influenza A virus (H1N1) strain A/PR/8/34 (PR8) infection. Quantitative analysis of viral *nucleoprotein (NP)* RNA levels in both cellular and supernatant samples (Fig. 2A, B), along with Western blot (Fig. 2C, D) and immunofluorescence detection (Fig. 2E, F) of NP protein, collectively demonstrated the ability of AmB to promote IAV infection. Moreover, a time-course analysis using RT-qPCR revealed that the potentiating effect of AmB was already evident at approximately 6 hpi (Fig. 2G). Furthermore, time-of-addition experiments conducted both in vitro and in vivo demonstrated that the enhancing effect of AmB is time-dependent, occurring only when AmB is administered during the early phase of infection (Fig. S6A–H). Collectively, these findings underscore its impact during the early phase of infection, suggesting that AmB may primarily influence viral entry.

This hypothesis was further supported by experiments using HIV/IAV (H5N1) pseudoviruses, which showed that AmB consistently augmented viral entry in multiple cell lines, including A549, HULEC-5a, 293T, and COS-7 cells, indicating that this effect is not restricted to a particular cell type (Fig. 2H). Additionally, nystatin—a structural analog of AmB—also promoted the entry of both HIV/IAV and HIV/SARS-CoV-2 pseudoviruses, further supporting a shared mechanism among this compound class (Fig. S7A–H). In light of previous reports that IFITM3 inhibits viral entry[22], and that AmB might counteract IFITM3-mediated

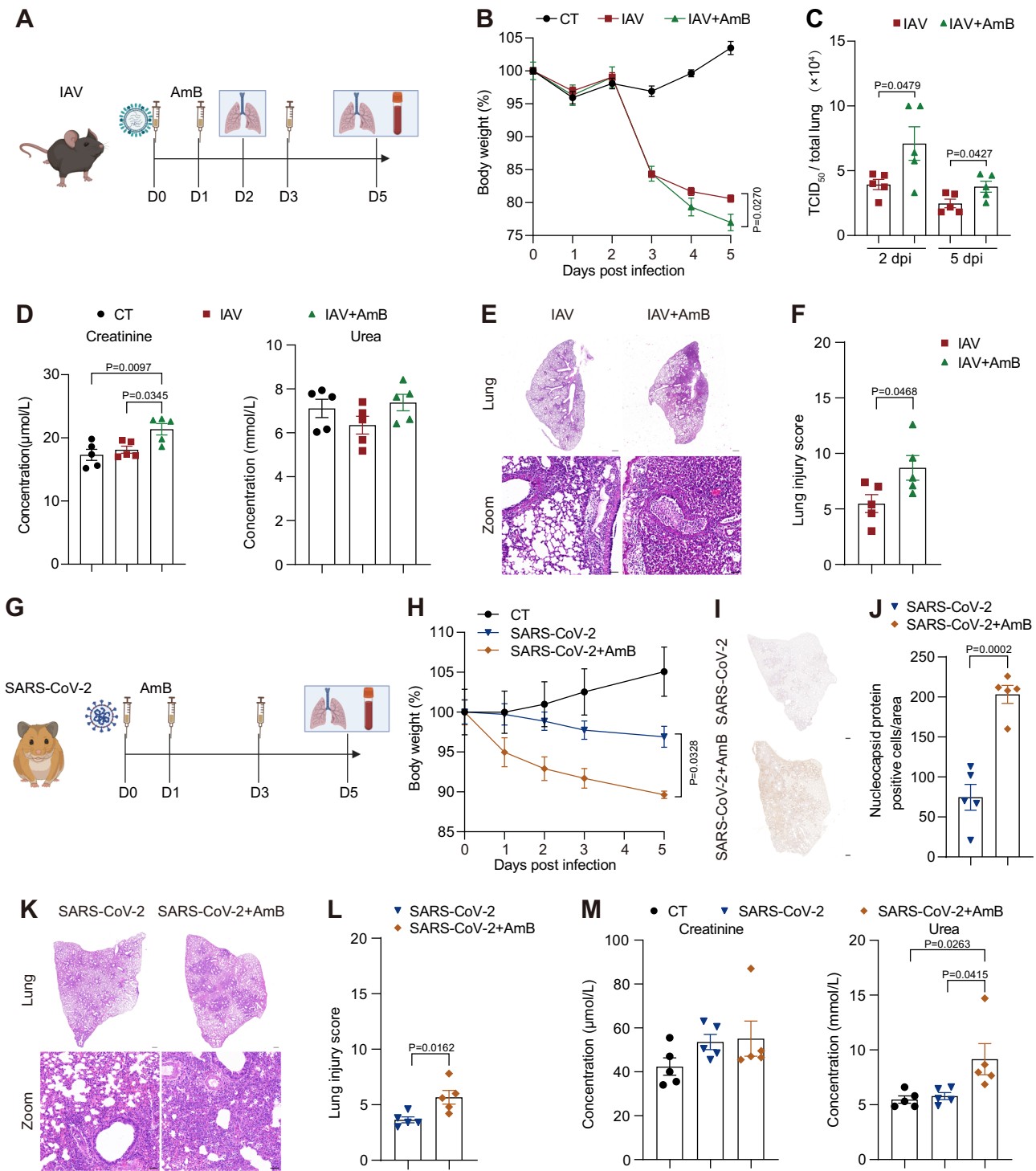

restriction[18], we assessed the baseline expression of *IFITM3* across these cell lines to evaluate its potential involvement (Fig. S8A). Interestingly, while A549 cells displayed low *IFITM3* expression and COS-7 cells exhibited undetectable levels, AmB still significantly enhanced HIV/IAV pseudoviruses entry in both cell types. To further substantiate the IFITM3-independent nature of this phenomenon, we conducted functional modulation experiments. siRNA-mediated *IFITM3* knockdown (Fig. S8B, C) and plasmid-based transient overexpression (Fig. S8D, E) only moderately modulated the entry-enhancing effects of AmB, indicating that other mechanisms are involved in this process.

We then aimed to pinpoint the specific step in the viral entry affected by AmB. To this end, we systematically evaluated the

sequential stages of viral entry, including receptor binding, endocytic internalization, endosomal acidification, membrane fusion, and nucleocapsid release. Firstly, RT-qPCR (Fig. 2I) and immunofluorescence analysis (Fig. 2J, K) revealed no difference in virus binding between AmB-treated and control cells. Similarly, there was no difference in the degree of internalization after attachment (Fig. 2L, M), ruling out the influence on endocytic uptake. Considering that IAV can exploit multiple endocytic routes, including clathrin-mediated endocytosis, macropinocytosis, and caveolin-dependent uptake[23,24], we used a panel of pathway-specific fluorescent probes to assess and found that AmB did not alter the efficiency of any of these major endocytic mechanisms (Figs. 2N, O, S9A–J). These findings collectively

**Fig. 1 | Amphotericin B (AmB) increases disease severity in influenza A virus (IAV)-infected mice and SARS-CoV-2-infected hamsters. A** Mice received an intraperitoneal injection of AmB (1 mg/kg), and 1 h later, they were inoculated intranasally with IAV strain A/PR/8/34 (200 PFU). AmB was readministered at 1 and 3 days post-infection (dpi). Lung tissues were harvested at 2 and 5 dpi, and serum was collected for analysis at 5 dpi. Created in BioRender. Li, S. (2026) https://BioRender.com/npcswrd. **B** Longitudinal monitoring of body weight in mice. Statistical significance is indicated for IAV+AmB versus IAV groups. **C** Infectious viral titers in lung homogenates were determined by 50% tissue culture infectious dose (TCID$_{50}$) assay at 2 and 5 dpi. **D** Renal function assessment in mice by measuring serum creatinine and urea levels at 5 dpi. **E, F** Representative hematoxylin and eosin (H&E)−stained lung sections are shown (**E**), along with pathological scoring for inflammation, hemorrhage, edema, atelectasis, and necrosis (**F**). Scale bars, 0.5 mm (grey), 50 μm (black). **G** Hamsters received the identical AmB treatment and dosing schedule as depicted in (**A**), followed by intranasal challenge with the SARS-CoV-2 Delta variant (10$^5$ TCID$_{50}$). Lung and serum samples were collected at 5 dpi. Created in BioRender. Li, S. (2026) https://BioRender.com/7oejn93. **H** Body weight changes in hamsters. Significant difference is indicated for SARS-CoV-2+AmB versus SARS-CoV-2 groups. **I, J** Lung sections were stained for the nucleocapsid protein (**I**). Nucleocapsid protein-positive cells were quantified per unit area (**J**). Scale bars, 0.5 mm. **K, L** H&E-stained lung sections (**K**) were assessed using the same scoring criteria as in (**F**). Scale bars, 0.5 mm (grey), 50 μm (black). **M** Assessment of renal function in hamsters by quantifying serum creatinine and urea levels at 5 dpi. Data are presented as mean ± SEM. $n = 5$ biologically independent animals per group (**B**−**F**; **H**−**M**). Statistical analysis was performed using two-way ANOVA followed by Tukey's multiple comparisons test (**B, H**), two-tailed unpaired Student's t-test (**C, F, J, L**), and one-way ANOVA followed by Tukey's multiple comparisons tests (**D, M**). Source data are provided as a Source Data file.

demonstrate that AmB enhances viral entry through mechanisms independent of initial binding or endocytosis.

## AmB facilitates IAV escape from late endosomes through enhanced membrane fusion

To further investigate the role of AmB in viral entry, we examined its effects on endosomal trafficking and membrane fusion dynamics. Under normal conditions, the sequential acidification of endosomal compartments induces conformational changes in the viral hemagglutinin protein, thereby triggering membrane fusion and the subsequent release of the nucleocapsid into the host cytosol[25,26].

Time-resolved co-localization analyses revealed distinct temporal dynamics of IAV intracellular trafficking in AmB-treated versus untreated cells. At 1–2 hpi, the distribution of IAV within early and late endosomes was similar between the two groups, indicating that AmB does not interfere with the initial internalization process (Fig. 3A–F). However, at 3 hpi, virions in AmB-treated cells exhibited significantly reduced retention within late endosomes and showed accelerated dispersal toward the nuclear region, implying an expedited escape from late endosomes (Fig. 3E, F).

To elucidate the mechanism by which AmB affects viral-endosomal membrane fusion, we implemented a dual fluorescent virus fusion assay. Purified PR8 virions were dual-labeled with the lipophilic dyes R18 (red) and SP-DiOC18 (green)[27,28]. Lipid mixing between the viral envelope and the host endosomal membrane was indicated by an increased intensity of green fluorescence, resulting from the release of self-quenching SP-DiOC18 and the dissolution of fluorescence resonance energy transfer (FRET). In cells without AmB treatment, predominant red fluorescence was observed at 3 hpi, whereas AmB-treated cells exhibited robust green fluorescence at the same time, confirming that AmB promotes viral-endosomal fusion (Fig. 3G, H). We next employed an acid-bypass assay. While AmB pretreatment promoted IAV infection under standard conditions, this enhancing effect was abolished when viral fusion was triggered directly at the plasma membrane using low-pH buffer, thereby bypassing endosomes (Fig. S10A). This demonstrates that AmB specifically facilitates endosomal membrane fusion, but not fusion at the plasma membrane. In parallel, our ratiometric pH measurements using LysoSensor™ revealed that AmB treatment induced a mild alkalinization of acidic organelles (Fig. S10B, C), consistent with previous reports[29]. Given that this modest pH elevation would typically be expected to impair acid-dependent fusion, the observed enhancement in membrane fusion suggests that AmB may facilitate viral entry through alternative mechanisms.

Consistent with accelerated endosomal escape, downstream analyses of nucleocapsid trafficking revealed enhanced nuclear accumulation of viral NP upon AmB treatment. Immunofluorescence quantification at 5 hpi showed a significant increase of viral NP in AmB-treated cells compared to controls (Fig. 3I, J). These findings were corroborated by subcellular fractionation coupled with RT-qPCR

analysis, which demonstrated that AmB treatment led to earlier and amplified viral replication (Fig. 3K, L).

To determine whether this effect was specific to late endosome-dependent viral entry, we employed additional pseudoviruses with different endosomal maturation requirements, including HIV/SARS-CoV-2 and HIV/MERS-CoV pseudoviruses that escape from late endosomes, as well as HIV/VSV pseudoviruses that escape from early endosomes[26,30–32]. In HeLa-hACE2 and Huh7 cells, pretreatment with AmB significantly enhanced the entry of HIV/SARS-CoV-2 and HIV/MERS-CoV pseudoviruses as quantified by luciferase assays (Fig. 4A, B). Conversely, the entry of the early endosome-dependent HIV/VSV pseudoviruses in A549 cells showed no enhancement and instead demonstrated a slight yet significant inhibition (Fig. 4C). In summary, our findings demonstrate that AmB selectively enhances the fusion between viral envelope and late endosome membranes, thereby accelerating viral escape and the nuclear delivery of the viral genome− underscoring the critical role of late endosomes in mediating this effect.

## AmB binds to GCase in the late endosome

Although previous studies have reported the rapid accumulation of AmB in endosomal compartments within 1 h[33], the functional relationship between AmB and endosomal components in the context of viral infection remained unclear. To elucidate this further, we developed a chemical proteomics approach (Fig. 4D). A biotinylated derivative of AmB (AmB-biotin) was synthesized (Figs. 4F, S11A−B) and confirmed to fully retain its potency in promoting virus entry (Fig. 4G). Late endosomes were isolated from A549 cells using sucrose density ultracentrifugation, and the purity of these fractions was validated by Western blot analysis (Fig. 4E). The AmB-biotin conjugate was then incubated with isolated late endosomes, allowing for the capture of interacting proteins using streptavidin magnetic beads. Subsequent SDS-PAGE and silver staining revealed that AmB-biotin specifically pulled down proteins with a molecular weight of ~60 kDa. Notably, the corresponding band was absent in the biotin-only control and appeared only faintly when excess free AmB was added (Fig. 4H). LC−MS/MS analysis of this band identified the protein as GCase (Fig. 4I). To further substantiate this interaction, we confirmed the specific enrichment of GCase in the AmB-biotin group, whereas negligible levels were detected in the control and competition samples (Fig. 4J). Moreover, surface plasmon resonance (SPR) analysis revealed a dose-dependent binding interaction between AmB and recombinant GCase, yielding a dissociation constant (K$_D$) of 23 μM (Fig. 4K). Collectively, these data establish GCase as a critical late endosomal target of AmB.

## AmB promotes viral entry via GCase activation and ceramide remodeling in late endosomes

GCase, the lysosomal hydrolase encoded by *GBA1*, plays a central role in glycosphingolipid metabolism by converting glucosylceramide to

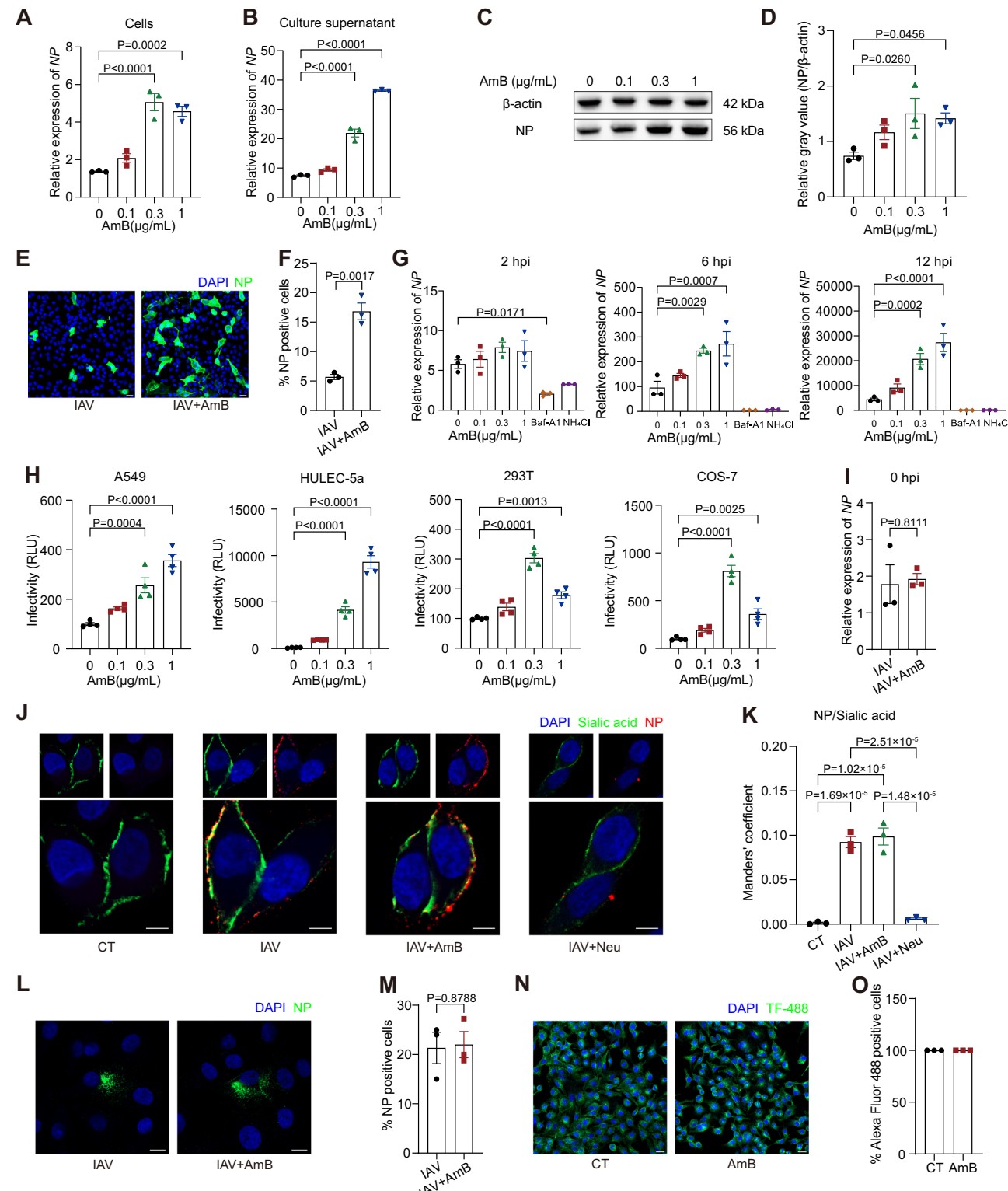

ceramide within late endosomes and lysosomes[34]. Given that the conical molecular structure of ceramide promotes negative membrane curvature and increases bilayer instability[35], the accumulation of ceramide may promote membrane fusion between the viral envelope and late endosomes. To illustrate the functional significance of the AmB-GCase interaction, we systematically evaluated whether AmB modulates GCase activity and alters ceramide levels in late endosomes during viral infection. Fluorometric assays revealed that AmB treatment significantly enhanced GCase activity across several cell lines

(Fig. 5A), with rapid, time-dependent activation kinetics that plateaued within 1 h in A549 cells (Fig. S12A). Complementary lipidomic analyses of late endosomes further revealed that AmB treatment led to a marked increase in ceramide levels during IAV infection (Fig. 5B, C).

To establish a causal link between GCase activation and the enhancement of viral entry mediated by AmB, we employed complementary approaches. We first confirmed that CRISPR–Cas9 knockout (KO) of *GBA1* attenuated the overall pro-infection effect of AmB in IAV infection (Figs. 5D, S12B). Next, to define the temporal

**Fig. 2 | Amphotericin B (AmB) enhances early-stage influenza A virus (IAV) infection by facilitating viral entry. A, B** A549 cells were pretreated with AmB 1 h before IAV infection (MOI = 0.1), and cells and supernatant were collected 24 h post-infection (hpi). Intracellular (**A**) and supernatant (**B**) *nucleoprotein (NP)* RNA levels were quantified by RT-qPCR. **C, D** NP protein levels in A549 cells treated and infected under the same conditions as described in (**A, B**) were evaluated by Western blot (**C**) and quantified as a relative gray value (**D**) at 24 hpi. **E, F** Immunofluorescence staining of NP (**E**) and quantification of NP-positive cells (**F**) were performed on A549 cells pretreated with 1 μg/mL AmB for 1 h, followed by IAV infection (MOI = 0.1) for 24 h. Scale bars, 20 μm. **G** Time-course analysis of intracellular *NP* RNA levels at 2, 6, and 12 hpi in A549 cells pretreated with the indicated concentrations of AmB, bafilomycin A1 (Baf-A1, 20 nM), or ammonium chloride (NH$_4$Cl, 10 mM) prior to IAV infection (MOI = 0.1). **H** A549, HULEC-5a, 293T, and COS-7 cells pretreated with AmB were infected with HIV/IAV pseudoviruses. Luciferase activity was measured 48 hpi to assess viral entry efficiency. **I** In the presence of 50 μg/mL cycloheximide (CHX), A549 cells pretreated with 1 μg/mL AmB for 1 h were incubated with IAV (MOI = 0.1) at 4 °C for 1 h. Following the removal of unbound virions by washing, *NP* RNA levels in the cells were quantified by RT-qPCR. **J, K** Binding assays were conducted with 50 μg/mL CHX present. Surface-bound virions were assessed under attachment conditions in four groups: untreated control (CT), IAV (MOI = 1), IAV with 1 μg/mL AmB pretreatment (IAV+AmB), and IAV following 50 mU/mL neuraminidase pretreatment as a negative control (IAV+Neu). Virions were visualized (**J**) and quantified (**K**) via NP and sialic acid co-staining. Colocalization was analyzed using the Manders' M2 coefficient (channel A: NP; channel B: sialic acid). Scale bars, 5 μm. **L, M** Following IAV (MOI = 1) attachment at 4 °C, A549 cells with or without 1 μg/mL AmB pretreatment were shifted to 37 °C for 2 h to allow internalization in the presence of 50 μg/mL CHX. Internalized virions were detected by NP immunostaining (**L**) and quantified (**M**). Scale bars, 10 μm. **N, O** A549 cells pretreated with 1 μg/mL AmB for 24 h were incubated with Alexa Fluor 488-conjugated transferrin (TF-488, 25 μg/mL) for 5 min at 37 °C. The internalized transferrin was then visualized by immunofluorescence (**N**), and the percentage of Alexa Fluor 488-positive cells was quantified (**O**). Scale bars, 20 μm. Data are presented as mean ± SEM. n = 3 independent experiments (**A–G, I–O**); n = 4 independent experiments (**H**). Statistical analysis was performed using one-way ANOVA followed by Dunnett's (**A, B, D, G, H**) or Tukey's (**K**) multiple comparisons tests, and two-tailed unpaired Student's t-test (**F, I, M, O**). Source data are provided as a Source Data file.

requirement for GCase activity, we performed time-of-addition experiments with the pharmacological GCase inhibitor conduritol B epoxide (CBE), used at a non-toxic concentration (Fig. S12C, D). CBE suppressed infection only when added during the early phase (Fig. 5E), mirroring the action window of AmB, whereas addition at later time points was ineffective (Fig. 5F). To directly test the role of GCase in viral entry, we measured pseudovirus entry. CBE treatment blunted the AmB-induced enhancement of entry for both HIV/IAV and HIV/SARS-CoV-2 pseudoviruses (Fig. 5G). Consistent with this, *GBA1* KO in 293T cells (Fig. 5H) and siRNA-mediated knockdown in A549 and HeLa-hACE2 cells (Figs. 5I, S12E) each substantially reduced the AmB-mediated enhancement of viral entry. Collectively, these data demonstrate that AmB promotes viral entry by activating GCase, leading to ceramide accumulation in late endosomes.

## AmB modulates endosomal composition via GCase-dependent RAB7 upregulation

To elucidate whether AmB also modulates endosomal protein composition, we performed a data-independent acquisition-based quantitative proteomic analysis on early and late endosomes isolated from A549 cells, pretreated with 1 μg/mL AmB for 1 h and subsequently infected with PR8 at an MOI of 1 for 4 h. Principal component analysis (PCA) revealed distinct segregation between the AmB-treated and untreated groups, indicating that AmB induces substantial remodeling of the endosomal proteome (Fig. 6A). Notably, several critical components of the endocytic pathway were significantly altered in the late endosomes of AmB-treated cells. While the late endosomal marker RAB7, caveolin-1 (CAV1), and RAB-interacting lysosomal protein (RILP) were significantly enriched, the levels of other endosomal proteins such as KXD1, MOSPD2, OSBPL11, and AP5S1 were reduced (Figs. 6B–D, S13A–J). In particular, the small GTPase RAB7, which is integral to endosomal maturation by regulating lysosomal fusion, vesicle trafficking, and membrane remodeling[36], exhibited a pronounced upregulation, as confirmed by Western blot analysis (Fig. 6E–I).

To clarify the influence of AmB-GCase interaction on late endosomal protein composition, we compared the protein expression profiles of late endosomes isolated from wild-type and *GBA1* KO 293T cells (Fig. 6J). Genetic ablation of *GBA1* resulted in reduced levels of NPC1 and RAB7, coupled with increased LAMP1 expression (Fig. 6K–O). This reciprocal regulatory pattern—wherein AmB treatment increases RAB7 levels while *GBA1* deletion decreases them—provides strong evidence that active GCase is necessary to sustain normal RAB7 levels and ensure proper endosomal maturation. These findings collectively underscore that AmB exerts its modulatory effects on endosomal protein composition through a GCase-dependent pathway.

Based on our comprehensive findings, we propose a mechanistic model in which AmB binding triggers a conformational activation of GCase. This activation leads to ceramide accumulation and a GCase-dependent upregulation of RAB7, which remodels endosomal architecture. Ultimately, these changes promote the fusion of the viral envelope with late endosomal membranes (Fig. 6P). Taken together, our findings identify GCase-regulated ceramide synthesis and endosomal maturation as novel pharmacological targets through which AmB modulates membrane fusion and enhances viral infection.

## Discussion

In this study, we address a critical therapeutic dilemma in managing patients with invasive pulmonary aspergillosis, where conventional antifungal treatments may inadvertently potentiate viral pathogenesis through previously undefined mechanisms. Through a comprehensive, multidisciplinary approach that combined in vitro cellular assays, animal models, and a retrospective clinical analysis, we demonstrated that AmB aggravates IAV and SARS-CoV-2 infections and their associated pathogenesis. Specifically, AmB promotes the membrane fusion of the viral envelope and late endosome membranes during viral entry. Mechanistically, AmB directly binds to GCase in the late endosomes, activating its enzymatic activity and elevating the ceramide levels, which may destabilize the lipid bilayers of these compartments. AmB also promotes the maturation of the late endosomes in a GCase-dependent manner, as demonstrated by elevated levels of late endosomal protein markers. Collectively, these coordinated alterations remodel the membranes of late endosomes, thereby facilitating viral entry by promoting membrane fusion.

Phenotypic analyses revealed that AmB specifically enhances late endosome-dependent viral entry without affecting early endocytic processes. This compartment-specific activity is demonstrated by its negligible effect on HIV/VSV entry, which relies on early endosomes, compared to the significant enhancement of IAV and SARS-CoV-2 entry that requires late endosomal maturation. This effect can be explained by the selective activation of GCase localized in late endosomes. Previous studies had attributed the antiviral effects of AmB to interference with cholesterol-dependent caveolin-mediated endocytosis, which reduced the infectivity of viruses like enterovirus 71 and Japanese encephalitis virus[14–16]. However, these models fail to explain the virus entry-promoting activity observed with IAV and coronaviruses. Our results demonstrate that the paradoxical effects of AmB stem from its ability to modulate host lipid metabolism rather than perturb early endocytosis. Notably, while prior studies have shown that nystatin—a structural congener of AmB—inhibits SARS-CoV-2 replication at post-

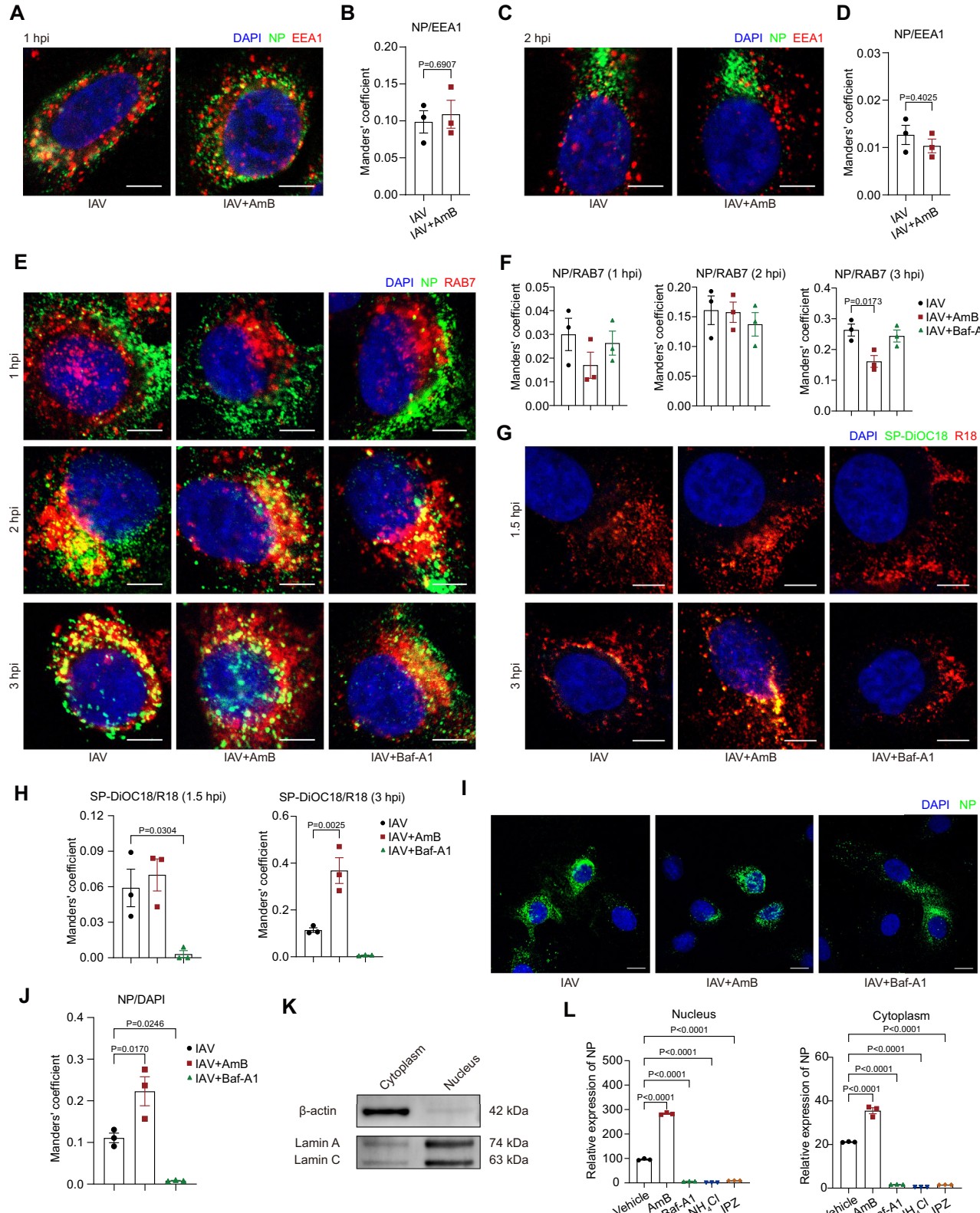

entry stages[21], our findings reveal its ability to enhance the entry of both IAV and SARS-CoV-2 pseudoviruses.

Whereas earlier investigations attributed the virus entry-promoting effects of AmB largely to IFITM3 interference[18], our systematic gain- and loss-of-function experiments unequivocally excluded IFITM3 as a major mediator. Instead, our unbiased pull-down assay identified GCase in the late endosomes as a protein binding partner of

AmB. It is interesting to note that this combination is theoretically reasonable as both AmB and GCase exhibit lipophilicity. Additionally, SPR experiments showed that AmB and GCase bind with moderate affinity ($K_D = 23\,\mu M$). The following mechanistic insights provide a partial explanation for the previously reported discrepancies regarding the role of host glycosphingolipid metabolism in viral pathogenesis. We found that AmB activates GCase and elevates ceramide levels

**Fig. 3 | Amphotericin B (AmB) promotes the fusion of influenza A virus (IAV) with late endosomes and enhances the nuclear import of viral genomes.** Unless otherwise specified, A549 cells were pretreated with 1 μg/mL AmB for 1 h. Where indicated, bafilomycin A1 (Baf-A1, 20 nM), ammonium chloride (NH$_4$Cl, 10 mM), or importazole (IPZ, 20 μM) were used as controls. Following pretreatment, cells were incubated with IAV strain A/PR/8/34 (MOI = 1) at 4 °C for 1 h, and shifted to 37 °C for the indicated times. All steps were conducted with 50 μg/mL cycloheximide (CHX) present. **A–D** Colocalization of nucleoprotein (NP) with early endosome antigen 1 (EEA1) was analyzed at 1 h post-infection (hpi) (**A**, **B**) and 2 hpi (**C**, **D**). Representative images (**A**, **C**) and Manders' M2 coefficient quantification (channel A: NP; channel B: EEA1) (**B**, **D**) are shown. Scale bars, 5 μm. **E**, **F** At the indicated times post-infection, cells were co-stained for NP and the late endosome marker RAB7 (**E**). NP–RAB7 colocalization was quantified using Manders' M2 coefficient (channel A: NP; channel B: RAB7) (**F**). Scale bars, 5 μm. **G**, **H** A549 cells were pretreated with 1 μg/mL AmB for 1 h in the presence of 50 μg/mL CHX. Cells were then infected with IAV dually labeled with R18 and SP-DiOC18. Representative images captured at 1.5 and 3 hpi are shown (**G**), and viral fusion was quantified using Manders' M2 coefficient (channel A: SP-DiOC18; channel B: R18) (**H**). Scale bars, 5 μm. **I**, **J** Nuclear import of NP was assessed at 5 hpi by immunofluorescence (**I**) and quantified based on NP-DAPI colocalization using Manders' M2 coefficient (channel A: NP; channel B: DAPI) (**J**). Scale bars, 10 μm. **K** Western blot analysis of the cytoplasmic marker β-actin and nuclear marker lamin A/C in the respective fractions, following standardization by BCA protein assay. Images are representative of three independent experiments. **L** NP RNA levels in nuclear and cytoplasmic fractions at 5 hpi were measured by RT-qPCR. Data are presented as mean ± SEM. n = 3 independent experiments (**A–J**, **L**). Statistical analysis was performed using two-tailed unpaired Student's t-test (**B**, **D**) and one-way ANOVA followed by Dunnett's multiple comparisons test (**F**, **H**, **J**, **L**). Source data are provided as a Source Data file.

in late endosomes, thereby altering the endosomal membrane composition to promote membrane fusion. The nonlinear dose-response relationships observed in modulating key enzymes highlight the complexity of glycosphingolipid dynamics. Notably, *GBA1* knockout leads to glucosylceramide accumulation that impedes IAV trafficking to late endosomes[37], while glucosylceramide synthase (GCS) deletion similarly inhibits infection through reduced glucosylceramide levels[38]. These findings suggest that optimal viral entry requires precise regulation of the glucosylceramide-to-ceramide ratio. In this context, AmB-induced activation of GCase may achieve an optimal balance in glycosphingolipid metabolism to promote membrane fusion. Comparable observations with other agents further support this model. For instance, fenretinide activates key enzymes in the ceramide synthesis pathway, thereby increasing cellular ceramide levels and enhancing IAV infection by more than 50%[39]. In contrast, despite elevating ceramide levels, exogenous acid sphingomyelinase simultaneously depletes plasma membrane sphingomyelin, ultimately inhibiting IAV entry[40]. Collectively, these findings underscore that viral entry efficiency is finely regulated by the balance of glycosphingolipid metabolism, where even subtle metabolic changes can either promote or inhibit viral entry.

Our findings carry significant clinical implications. Retrospective analysis suggests that the systemic use of AmB was associated with a significantly higher incidence of subsequent viral infection compared to other antifungal therapies, which was further supported by multivariable analysis confirming AmB as an independent risk factor. These results raise an urgent need to reassess current antifungal guidelines for managing IPA infections. Notably, our data indicate that even clinically standard doses of AmB (1 mg/kg) can potentiate viral infection, while higher doses induce unacceptable toxicity. This suggests a narrow therapeutic window for achieving effective fungal eradication without exacerbating viral pathogenesis. To minimize the risk of AmB-associated secondary viral infection, it may be prudent to avoid its use, when possible, during primary IPA infection as well as in cases of viral-IPA co-infection. Given that AmB administration during the late stage of viral infection does not exacerbate the viral infection, its use for IPA that complicates late-stage viral infection is theoretically safe but still requires clinical validation. For other antifungal agents, our findings demonstrate that itraconazole and posaconazole have a certain inhibitory effect on pseudovirus entry. This observation aligns with preclinical evidence supporting the use of azoles, such as itraconazole, as safer alternatives, demonstrating both antifungal efficacy and antiviral synergy[41,42].

Several limitations of our study warrant consideration. First, while our clinical cohort represents the largest analysis of subsequent viral infection following AmB treatment to date, its retrospective design entails inherent limitations. Despite adjustments via propensity score matching and multivariate analysis, residual unmeasured confounding could influence the outcomes. Therefore, prospective cohort studies are required to substantiate these findings. Second, while our animal

models employed dosing regimens that approximate clinical usage, the impact of different AmB formulations (such as liposomal versus deoxycholate AmB) on viral pathogenesis remains to be explicitly evaluated. Moreover, the mechanisms through which GCase downregulation leads to decreased RAB7 expression remain unclear and warrant further study.

In summary, our study substantially advances our understanding of the pharmacological crosstalk between antifungal agents and viral infections, demonstrating that off-target metabolic perturbations—specifically, AmB-driven GCase activation and ceramide remodeling—significantly influence viral pathogenesis. To translate these insights into clinical innovation, we advocate conducting high-resolution structural studies of the AmB-GCase complex, which will facilitate the rational design of next-generation antifungals with minimized off-target effects. In addition, targeting GCase through allosteric modulation may provide a novel antiviral strategy. Since our study primarily focused on IAV and SARS-CoV-2, the identified GCase-ceramide axis likely extends to other late endosome-dependent pathogens, such as Ebola and Lassa fever viruses, highlighting its potential as a broad-spectrum antiviral target that warrants further virological validation. Collectively, this work underscores the need to redefine antimicrobial stewardship in the era of co-epidemics, where therapeutic interventions must be evaluated not only for their direct microbicidal potency but also for their broader influence on host-pathogen interactions.

## Methods

### Cells

The cell lines used in this study included A549 (Cat# CCL-185), HULEC-5a (Cat# CRL-3244), 293T (Cat# CRL-3216), COS-7 (Cat# CRL-1651), HeLa (Cat# CCL-2), MDCK (Cat# CCL-34), and Vero E6 (Cat# CRL-1586) cells, all obtained from the American Type Culture Collection (ATCC, USA). Huh7 cells were kindly provided by Dr. Xuanling Shi (Tsinghua University). A CRISPR-Cas9-mediated *GBA1* knockout 293T cell line (Cat# SY-KO-00256) was acquired from Cyagen Biosciences (China). Stable hACE2-expressing cell lines (HeLa-hACE2 and A549-hACE2 cells) were generated in-house via retroviral transduction[43]. Briefly, the full-length coding sequence of wild-type human ACE2 was amplified by PCR and subcloned into the pBabe-puro retroviral vector. The resulting construct was used to produce retroviruses to transduce the parental cells. Following transduction, cells were selected with medium containing 10 μg/mL puromycin (Sigma-Aldrich, USA) for 7 days to establish stable populations.

All cell lines, except HULEC-5a cells, were maintained in Dulbecco's Modified Eagle Medium (DMEM, Gibco, USA) supplemented with 10% fetal bovine serum (FBS, Gibco, USA), 100 IU/mL penicillin and 100 μg/mL streptomycin (Gibco, USA). HULEC-5a cells were cultured in MCDB 131 medium (Gibco, USA) supplemented with 10% FBS (Gibco, USA), 10 mM L-glutamine (Sigma-Aldrich, USA), 10 ng/mL recombinant human epidermal growth factor (Gibco, USA), and 1 μg/mL hydrocortisone (Sigma-Aldrich, USA), along with the same concentrations of

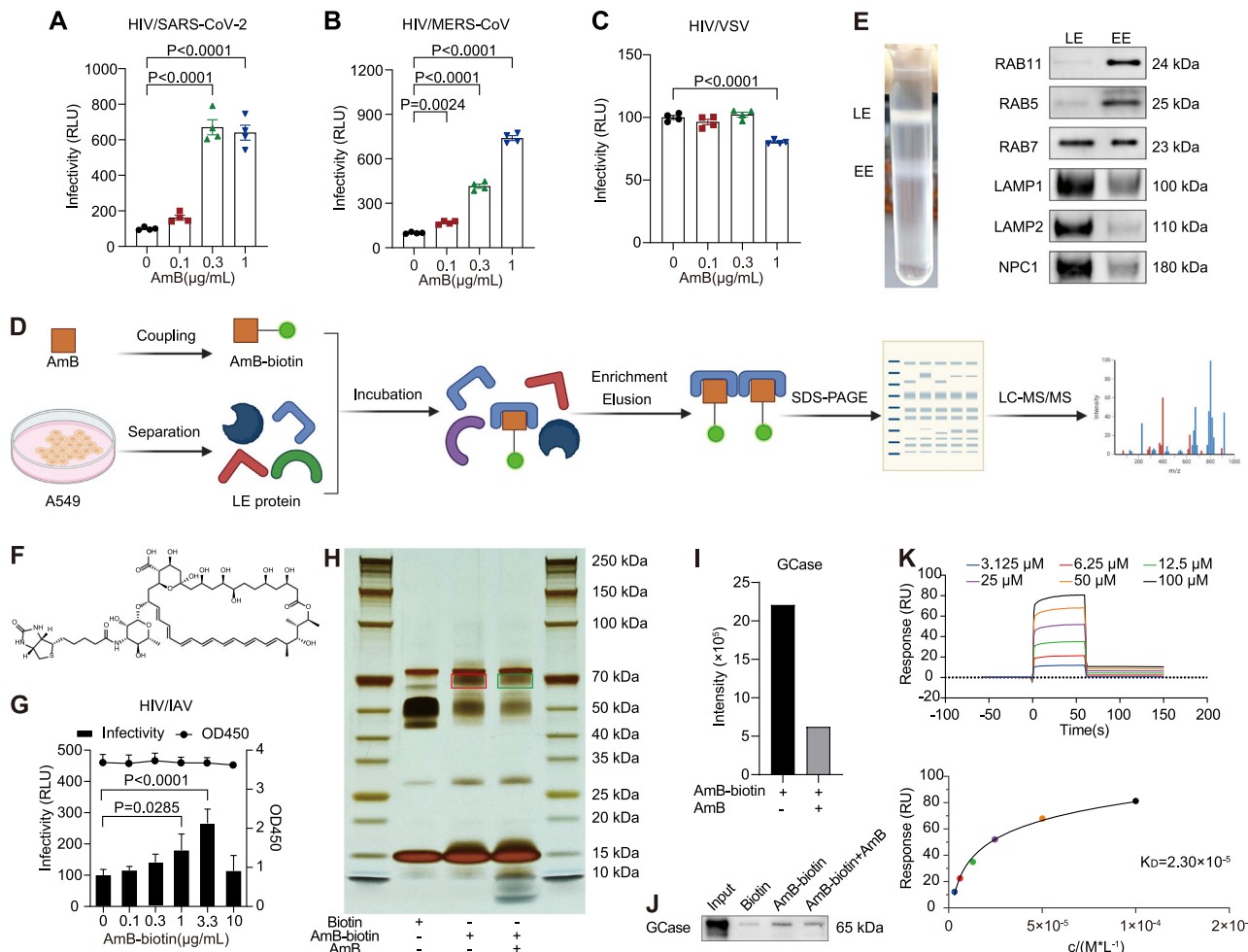

**Fig. 4 | Amphotericin B (AmB) selectively enhances viral entry via late endosomes and targets glucocerebrosidase (GCase) within these compartments.**
**A–C** HeLa-hACE2 (**A**), Huh7 (**B**), and A549 (**C**) cells were pretreated with the indicated concentrations of AmB for 1 h and subsequently infected with HIV/SARS-CoV-2, HIV/MERS-CoV, or HIV/VSV pseudoviruses, respectively. Viral entry was quantified by luciferase assay at 48 h post-infection (hpi). **D** Schematic workflow for identifying AmB-binding targets within late endosomes (LE). Biotinylated AmB (AmB-biotin) was incubated with purified LE. The resulting complexes were captured using streptavidin magnetic beads, eluted, separated by SDS-PAGE, and visualized via silver staining. Differential bands were subsequently excised for LC–MS/MS analysis. Created in BioRender. Li, S. (2026) https://BioRender.com/ilvgblu. **E** Validation of endosomal subpopulation isolation. Early endosomes (EE) and LE from A549 cells were separated by sucrose density ultracentrifugation, and the purity was confirmed with equal loading ensured by the BCA assay via immunoblotting with specific early and late endosomal markers. Images are representative of three independent experiments. **F** Chemical structure of the AmB-biotin conjugate. **G** Validation of AmB-biotin functionality. A549 cells pretreated with AmB-biotin were infected with HIV/IAV pseudoviruses, and infectivity was measured by luciferase assay at 48 hpi. **H** Silver staining analysis of the competitive pull-down assay. Three experimental conditions were evaluated: (1) biotin control with LE; (2) AmB-biotin with LE; and (3) AmB-biotin with LE in the presence of excess free AmB. Proteins were resolved by SDS-PAGE. The red and green boxes in the panel denote specific gel regions from lanes 2 and 3 that were excised for LC–MS/MS analysis, corresponding to differential bands present in lane 2, diminished in lane 3, and undetectable in lane 1. Images are representative of two independent experiments. **I** LC–MS/MS analysis was performed to quantify GCase levels in the different treatment groups. **J** Immunoblot analysis of GCase levels in pull-down samples from cell lysates. Images are representative of two independent experiments. **K** SPR analysis to assess the interaction between AmB and GCase. The determined equilibrium dissociation constant ($K_D$) is indicated in the panel. Data are presented as mean ± SEM. $n = 4$ independent experiments (**A–C**, **G**). Statistical analysis was performed using one-way ANOVA followed by Dunnett's multiple comparisons test (**A–C**, **G**). Source data are provided as a Source Data file.

penicillin and streptomycin as the other cell lines. All cultures were incubated at 37 °C in a humidified atmosphere containing 5% $CO_2$.

## Viruses

The influenza A virus (H1N1) strain A/PR/8/34 (PR8; Cat# VR-1469, ATCC, USA) was propagated in 9- to 11-day-old specific-pathogen-free (SPF) embryonated chicken eggs for 48 h at 37 °C. The eggs were then chilled at 4 °C for at least 12 h to restrict vascular bleeding. Allantoic fluid was harvested under biosafety level 2 (BSL-2) conditions, clarified by centrifugation at 3000 × $g$ for 10 min at 4 °C, and stored at −80 °C. Viral titers were determined using a plaque assay on MDCK cells. The SARS-CoV-2 strain (preservation number: CCPM-B-V-049-2207-20,

SARS-CoV-2/human/CHN/Delta-1/2021) was obtained from the National Pathogen Resource Center and propagated in Vero E6 cells. In infected hamsters, lung homogenates were subjected to virus quantification in Vero E6 cells using an endpoint titration method, with titers determined by a standard 50% tissue culture infectious dose ($TCID_{50}$) assay.

## In vivo experiments

SPF 6–8-week-old male C57BL/6 J mice and LVG Golden Syrian hamsters were obtained from Beijing Vital River Laboratory Animal Technology Co., Ltd. (China). Male animals were selected because previous studies have reported that males exhibit greater susceptibility and more severe disease phenotypes following respiratory viral infections

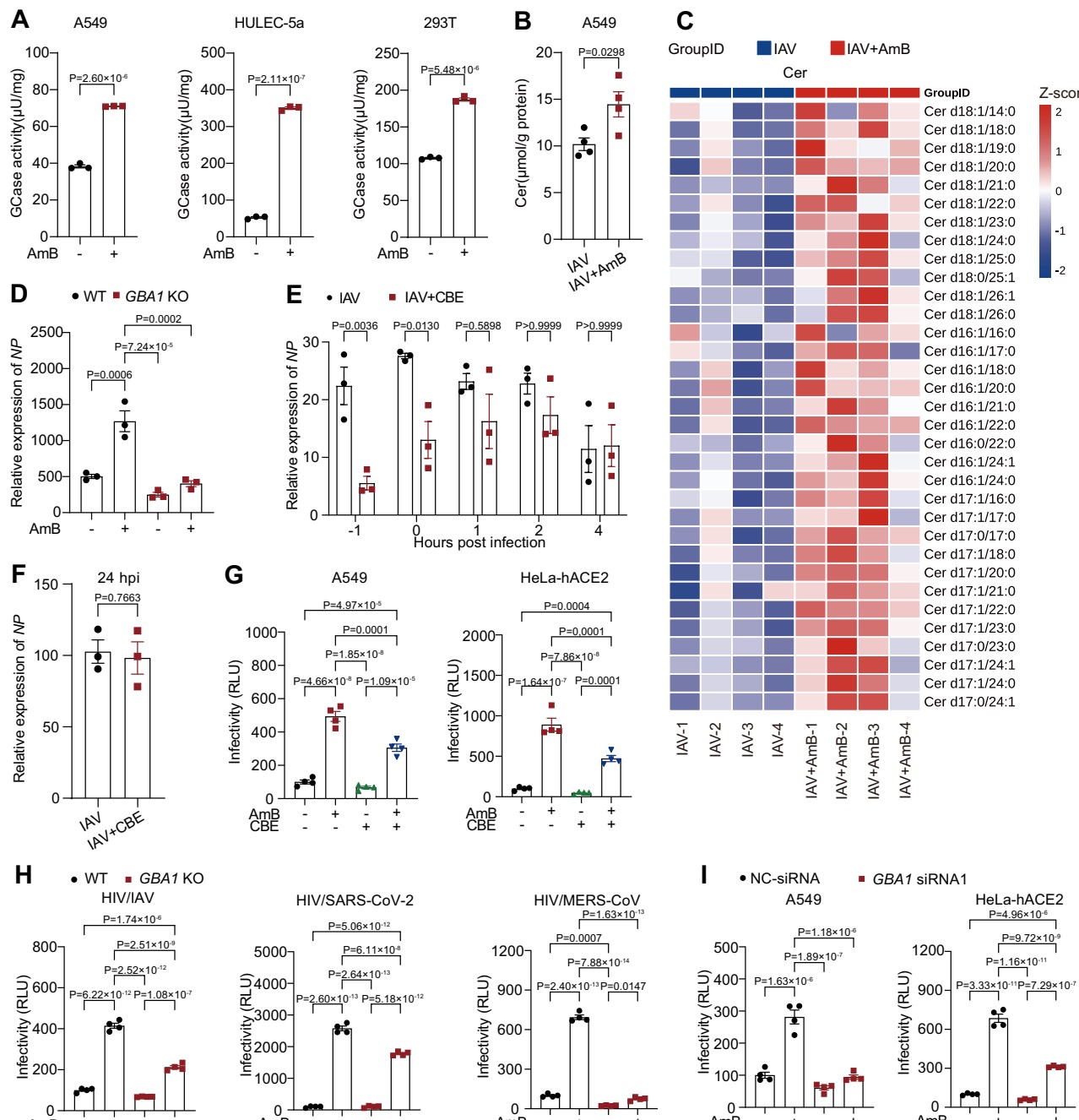

**Fig. 5 | Amphotericin B (AmB) enhances viral entry through glucocerebrosidase (GCase)-dependent ceramide production. A** GCase activity was determined using a fluorogenic substrate assay in cell lysates from A549 and HULEC-5a cells treated with 1 μg/mL AmB, and from 293T cells treated with 0.3 μg/mL AmB for 4 h. **B, C** Total ceramide (Cer) levels were quantified in purified late endosomes from A549 cells pretreated with 1 μg/mL AmB for 1 h and subsequently infected with influenza A virus (IAV, MOI = 1) for 4 h (**B**). The relative levels of individual ceramide subspecies across different treatment groups are shown, normalized by Z-score (**C**). **D** Wild-type (WT) and *GBA1* knockout (KO) 293T cells were pretreated with 0.3 μg/mL AmB and infected with IAV (MOI = 0.1) for 24 h. *Nucleoprotein (NP)* RNA levels were measured by RT-qPCR. **E** In the presence of 50 μg/mL cycloheximide (CHX), A549 cells were infected with IAV (MOI = 0.1). At −1, 0, +1, +2, or +4 h relative to infection, 100 μM conduritol B epoxide (CBE) was added, and samples were collected at 6 h post-infection (hpi) for RT-qPCR analysis of *NP* RNA. **F** A549 cells were first infected with IAV (MOI = 0.1), then treated with 100 μM CBE at 6 hpi. *NP* RNA levels were measured at 24 hpi. **G** A549 cells infected with HIV/IAV pseudoviruses and HeLa-hACE2 cells infected with HIV/SARS-CoV-2 pseudoviruses were pretreated with vehicle, 100 μM CBE, 1 μg/mL AmB, or both. Viral infectivity was assessed by luciferase assays at 48 hpi. **H** WT and *GBA1* KO 293T cells were treated with 0.3 μg/mL AmB and infected with the indicated pseudoviruses. Viral infectivity was assessed by luciferase assays at 48 hpi. **I** A549 and HeLa-hACE2 cells transfected with either negative control siRNA (NC-siRNA) or *GBA1*-targeting siRNA (*GBA1*-siRNA1) were pretreated with 1 μg/mL AmB. Subsequently, A549 cells were infected with HIV/IAV pseudoviruses and HeLa-hACE2 cells with HIV/SARS-CoV-2 pseudoviruses. Viral entry was then evaluated by luciferase assays at 48 hpi. Data are presented as mean ± SEM. $n = 3$ independent experiments (**A**, **D**–**F**); $n = 4$ independent experiments (**B**–**C**, **G**–**I**). Statistical analysis was performed using two-tailed unpaired Student's t-test (**A**–**B**, **F**), one-way ANOVA followed by Tukey's multiple comparisons test (**D**, **G**–**I**), and two-way ANOVA followed by Bonferroni's multiple comparisons test (**E**). Source data are provided as a Source Data file.

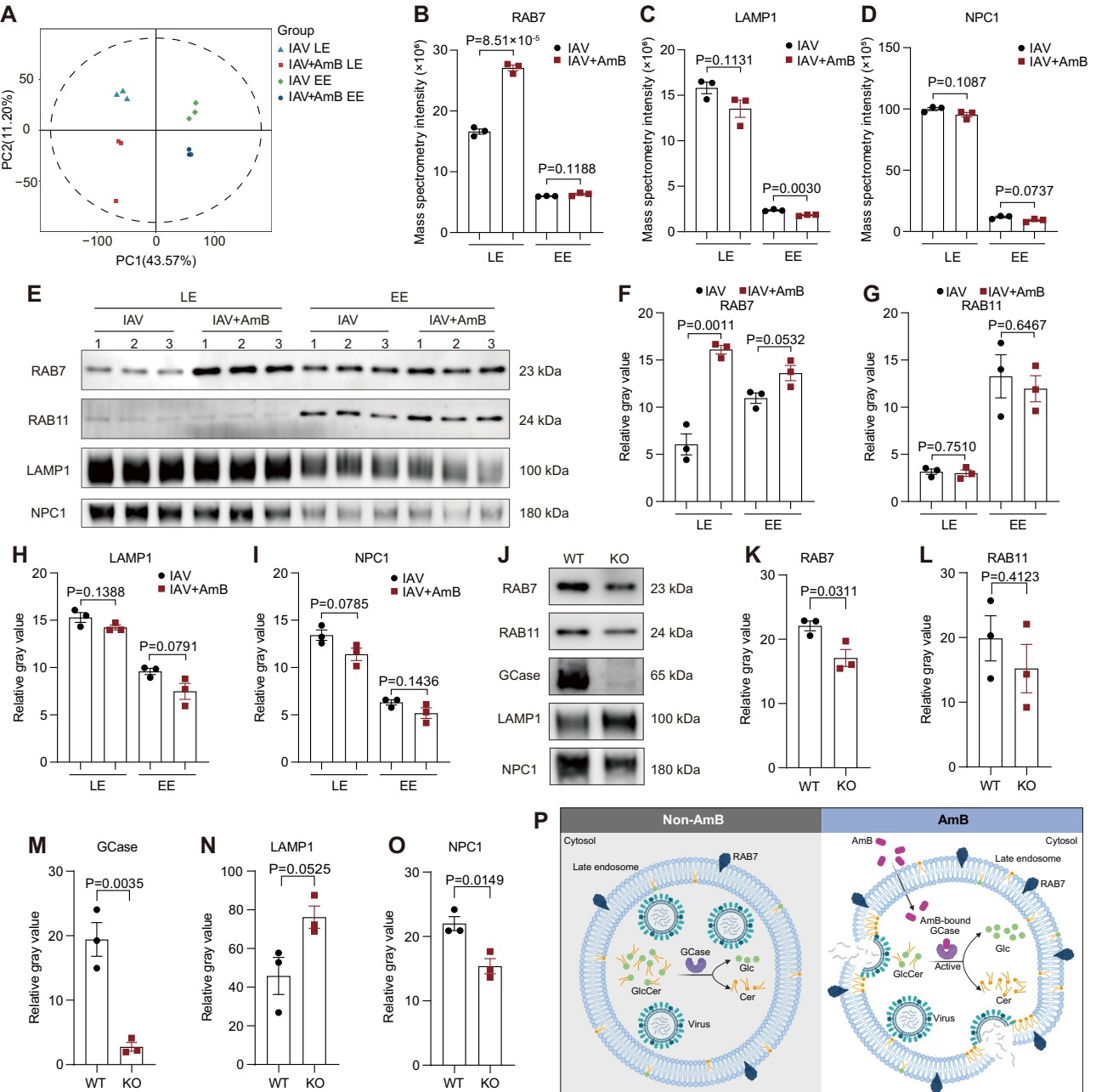

**Fig. 6 | Amphotericin B (AmB) modulates endosomal protein composition and maturation through glucocerebrosidase (GCase). A** A549 cells were pretreated with 1 µg/mL AmB for 1 h, and subsequently infected with influenza A virus (IAV, MOI = 1) for 4 h. Early endosomes (EE) and late endosomes (LE) from both groups were isolated and analyzed using data-independent acquisition-based quantitative proteomics. The principal component analysis (PCA) was performed.
**B–D** Proteomic profiling via mass spectrometry quantified the levels of RAB7 (**B**), LAMP1 (**C**), and NPC1 (**D**) under the experimental conditions outlined in (**A**).
**E–I** Endosomal fractions isolated as described in (**A**) were subjected to immunoblot analysis after normalization by BCA assay (**E**), followed by quantification of RAB7 (**F**), RAB11 (**G**), LAMP1 (**H**), and NPC1 (**I**) levels across treatment groups. **J–O** Immunoblot analysis was performed with equal loading ensured by BCA assay on late endosomes isolated from wild-type (WT) and *GBA1* knockout (KO) 293T cells (**J**). Quantitative

analyses of RAB7 (**K**), RAB11 (**L**), GCase (**M**), LAMP1 (**N**), and NPC1 (**O**) are presented. **P** The proposed mechanistic model delineates how AmB-mediated GCase binding perturbs glycolipid homeostasis to potentiate viral entry. Specifically, AmB binding may induce a conformational change in the active site of GCase, markedly enhancing its catalytic efficiency in converting glucosylceramide (GlcCer) to ceramide (Cer) and glucose (Glc). This metabolic shift promotes fusion via two distinct mechanisms: (1) ceramide-enriched microdomains alter membrane rigidity and curvature to create fusion-permissive regions, and (2) dysregulated glycolipid ratios influence RAB7-mediated endosomal maturation, further enhancing fusion competence. Created in BioRender. Li, S. (2026) https://BioRender.com/85sile9. Data are presented as mean ± SEM. *n* = 3 independent experiments (**A–O**). Statistical analysis was performed using two-tailed unpaired Student's t-test (**B–D**, **F–I**, **K–O**). Source data are provided as a Source Data file.

compared to females[44,45], which is consistent with our preliminary observations. The animals were housed five per cage with ad libitum access to food and water at a temperature of $24 ± 2 °C$, relative humidity of $50 ± 5\%$, and a 12 h light/12 h dark cycle. They were acclimated to the facility for 5 days prior to the experiments.

To assess AmB toxicity, healthy animals received intraperitoneal injections of AmB at doses of 0, 0.2, 0.5, 1, or 3 mg/kg administered at 0, 24, and 72 h, with PBS-treated animals serving as controls. Body weight and survival were monitored daily. At 5 days post-injection, blood samples were collected via cardiac puncture, and lung, liver, and

kidney tissues were harvested. Blood samples were processed for routine hematological and biochemical analyses by Beijing Yiming Fuxing Biotechnology Co., Ltd. (China), while tissues were fixed in 4% paraformaldehyde (PFA, Beyotime, China) for histopathological evaluation using hematoxylin and eosin (H&E) staining.

For infection studies, mice and hamsters were challenged with PR8 and SARS-CoV-2, respectively. Both groups of infected animals received AmB under the AmB-pre regimen, defined as an intraperitoneal dose of 1 mg/kg administered 1 h before challenge, followed by additional doses at 24 and 72 h post-infection (hpi). Another independent group of IAV-infected mice received AmB under the AmB-post regimen, with administration initiated at 48 hpi and repeated at 72 and 96 hpi. Animals were anesthetized with 50 mg/kg of 1% pentobarbital sodium administered intraperitoneally. Mice were inoculated intranasally with 200 plaque-forming units (PFU) of PR8 in 30 μL PBS, while hamsters received $10^5$ TCID$_{50}$ of SARS-CoV-2 in 0.1 mL PBS via the same route. PBS-inoculated animals served as uninfected controls. Body weight and survival were recorded, and at designated time points, animals were euthanized for the collection of blood samples and lung tissues for subsequent virological and pathological analyses. Infectious viral titers in lung homogenates from IAV-infected mice were determined by TCID$_{50}$ assay on MDCK cells.

## Lung pathology

Lung tissue samples were fixed in 4% PFA (Beyotime, China) for 24 h and subsequently dehydrated using a fully automated tissue processor (Sakura, USA) before paraffin embedding (Leica Biosystems, Germany). Paraffin-embedded tissues were sectioned at 5 μm using a microtome (Leica Biosystems, Germany) and stained with an H&E staining system (Leica Biosystems, Germany). Whole-slide images were acquired using a Pannoramic MIDI II Digital Scanner (3DHISTECH, Hungary) and visualized using SlideViewer software version 2.6.0 (3DHISTECH, Hungary). Histopathological assessment was conducted using a semi-quantitative scoring system[46]. Briefly, H&E-stained sections were evaluated to quantify microscopic changes in lung architecture, including alveolar and interstitial inflammation, hemorrhage, edema, atelectasis, and necrosis. Each variable was graded on a five-point scale: no injury (0 points), damage affecting 25% (1 point), 50% (2 points), 75% (3 points), and diffuse damage (4 points). Scoring was performed independently by two blinded experimenters, and the average score was used for analysis.

## Immunohistochemistry

Lung sections from SARS-CoV-2-infected hamsters were deparaffinized in xylene and subsequently rehydrated through a graded ethanol series. Antigen retrieval was performed in 10 mM sodium citrate buffer (pH 6.0) by microwave heating for 20 min. Endogenous peroxidase activity was quenched with 0.3% $H_2O_2$ for 15 min, and non-specific binding was blocked with normal serum for 30 min. After blocking, the sections were washed briefly in PBS and then incubated overnight at 4 °C with a primary antibody against the SARS-CoV-2 nucleocapsid protein (Cat# 40143-R001, dilution 1:500; Sino Biological, China). Following three 5-min washes with PBS, the sections were incubated with a horseradish peroxidase (HRP)-conjugated anti-rabbit IgG secondary antibody (Cat# SE134, dilution 1:200; Solarbio, China) for 1 h at room temperature. After another three 5-min washes, immunoreactivity was visualized using diaminobenzidine (DAB) staining, and the nuclei were counterstained with hematoxylin. After dehydration and mounting, whole-slide images were acquired using a Pannoramic MIDI II Digital Scanner (3DHISTECH, Hungary) and visualized with SlideViewer software version 2.6.0 (3DHISTECH, Hungary). Unless otherwise specified, other immunohistochemistry-related reagents were purchased from Zsbio (China).

## Retrospective cohort study

We conducted a retrospective cohort study using the electronic medical records of adult inpatients at China-Japan Friendship Hospital between September 2016 and September 2025. Patients were included if they had a positive *Aspergillus* culture from a respiratory sample and received systemic antifungal therapy. Patients with insufficient medical records were excluded. Based on treatment, patients were classified into two groups: the AmB group was defined by the administration of systemic AmB for ≥3 consecutive days, and the Non-AmB group by the absence of such treatment.

## Plasmids

Plasmid constructs were generated using standard molecular cloning techniques. The expression vectors for the spike (S) proteins of SARS-CoV-2 and MERS-CoV, as well as for VSV-G, were previously established in our laboratory[43,47] and were constructed by inserting the respective full-length cDNA sequences into the pcDNA3.1(+) backbone. The pcDNA-IFITM3 expression plasmid was constructed by inserting the human *IFITM3* gene into the *BamHI/EcoRI* sites of the pcDNA3.1(+) vector; this construct was custom-synthesized and verified by Beijing Ruibiotech Co., Ltd. (China). The CMV/R-HA and CMV/R-N1NA expression plasmids, corresponding to influenza A virus (H5N1) strains A/Anhui/1/2005 and A/Thailand/1(KAN-1)/2004, respectively, were kindly provided by Dr. Linqi Zhang and Dr. Xuanling Shi (Tsinghua University).

## Pseudovirus infection assay

To generate HIV-based pseudoviruses, $4 \times 10^6$ 293T cells were co-transfected with 10 μg of pNL-Luc E- R- and 10 μg of the virus envelope protein expression vectors. The medium was replaced with fresh medium 12 h post-transfection, and supernatants were collected 48 h later and filtered through a 0.45 μm filter (Merck Millipore, USA). Pseudovirus preparations were normalized by quantifying the p24 antigen using an Alliance™ HIV-1 p24 Antigen ELISA Kit (Cat# NEK050, Revvity, USA). For infection assays, the supernatant containing 5 ng pseudotyped virus (p24) was added to 96-well plates seeded with 7500 cells per well. At 48 h post-infection, cells were lysed using Luciferase Cell Culture Lysis Reagent (Cat# E1531, Promega, USA). Then, 10 μL of lysate was mixed with 50 μL of Luciferase Assay System substrate (Cat# E4550, Promega, USA) for luminescence measurement. Data were acquired using a Spark® multimode microplate reader running SparkControl™ software version 2.2 (Tecan, Switzerland).

## In vitro pharmacological activity assay

Cells were pretreated for 1 h at 37 °C prior to infection with the following compounds from MedChemExpress, used at the indicated concentrations: voriconazole (Cat# HY-76200; 0.1, 1, 10, and 100 μg/mL), isavuconazole (Cat# HY-14273; 0.1, 1, 10, and 100 μg/mL), posaconazole (Cat# HY-17373; 0.1, 1, 10, and 100 μg/mL), itraconazole (Cat# HY-17514; 0.1, 1, 10, and 100 μg/mL), caspofungin (Cat# HY-17006; 0.1, 1, 10, and 100 μg/mL), amphotericin B (AmB, Cat# HY-B0221; 0.1, 0.3, and 1 μg/mL), bafilomycin A1 (Baf-A1, Cat# HY-100558; 20 nM), ammonium chloride (NH$_4$Cl, Cat# HY-Y1269; 10 mM), nystatin (Cat# HY-17409; 1, 3.3, 10, and 33 μg/mL), importazole (IPZ, Cat# HY-101091; 20 μM), or conduritol B epoxide (CBE, Cat# HY-100944; 100 μM).

## Cell viability assay

Cell viability was evaluated using the Cell Counting Kit-8 (CCK-8, Cat# CK04, Dojindo Laboratories, Japan). Briefly, 7,500 cells were seeded per well in 96-well plates and allowed to adhere for 24 h. Following treatment with the specified compounds for 48 h, 10 μL of CCK-8 reagent was added to each well, and the plates were incubated for 2 h at 37 °C. Absorbance at 450 nm was measured using a Spark® multimode microplate reader running SparkControl™ software version 2.2 (Tecan, Switzerland) to determine cell viability.

## RNA interference

Prior to transfection, A549 cells were seeded and grown to ~60% confluency. At 24 h post-seeding, the cells were transfected with siRNAs targeting *IFITM3* or *GBA1*. For *IFITM3* silencing, cells were transfected with a validated siRNA (Cat# s195035, Thermo Fisher Scientific, USA) or a non-targeting control siRNA (Cat# 4390844, Thermo Fisher Scientific, USA) using Lipofectamine RNAiMAX Transfection Reagent (Cat# 13778150, Thermo Fisher Scientific, USA), following the manufacturer's instructions. For *GBA1* knockdown, four distinct siRNA duplexes and a negative control siRNA (NC-siRNA) were obtained from GenePharma (China) and delivered using the siRNA-Mate™ transfection reagent (Cat# 04027, GenePharma, China). The sequences of the siRNAs targeting *GBA1* are provided in Table S5. At 48 h post-transfection, the cells were either harvested for RNA extraction and RT-qPCR to verify gene silencing or subjected to pseudovirus infection assays.

## RNA extraction and RT-qPCR

Total RNA from cells and tissues was extracted using TRIzol reagent (Cat# 15596018CN, Thermo Fisher Scientific, USA), while RNA from cell culture supernatants was purified with the QIAamp Viral RNA Mini Kit (Cat# 52904, QIAGEN, Germany), according to the manufacturers' protocols. RNA concentration and purity were assessed by measuring absorbance ratios using a NanoPhotometer® NP80 spectrophotometer (Implen, Germany). For cDNA synthesis, 1 μg of total RNA was reverse-transcribed with the Hifair® III 1st Strand cDNA Synthesis SuperMix (Cat# 11141ES10, Yeasen, China). Quantitative PCR was subsequently performed using PowerUp™ SYBR™ Green Master Mix (Cat# A25742, Thermo Fisher Scientific, USA) on a QuantStudio™ 5 Real-Time PCR System running QuantStudio™ Design & Analysis Software version 1.5.1 (Thermo Fisher Scientific, USA). The resulting mRNA levels were normalized to the endogenous control *GAPDH*. All primers were synthesized by Beijing Ruibiotech Co., Ltd. (China), and their sequences are provided in Table S6.

## Western blotting

Protein concentrations were determined using the Pierce™ BCA Protein Assay Kit (Cat# 23225, Thermo Fisher Scientific, USA). Proteins were denatured at 100 °C for 10 min, resolved on 4–20% gradient SDS-PAGE gels (ACE Biotechnology, China), and transferred to poly-vinylidene fluoride membranes (Merck Millipore, USA). The membranes were blocked in 5% non-fat milk in TBST for 1 h at room temperature and incubated overnight at 4 °C with primary antibodies: mouse anti-IAV NP (Cat# ab20343, 1:2,000; Abcam, USA), mouse anti-β-actin (Cat# A5316, 1:1,000; Sigma-Aldrich, USA), mouse anti-Lamin A/C (Cat# 4777S, 1:2,000; Cell Signaling Technology, USA), rabbit anti-RAB5A (Cat# 2143, 1:1,000; Cell Signaling Technology, USA), rabbit anti-RAB11 (Cat# 15903-1-AP, 1:1,000; Proteintech, USA), rabbit anti-RAB7 (Cat# ab137029, 1:1,000; Abcam, USA), rabbit anti-LAMP1 (Cat# 9091S, 1:1,000; Cell Signaling Technology, USA), rabbit anti-LAMP2 (Cat# PA1-655, 1:1,000; Thermo Fisher Scientific, USA), rabbit anti-NPC1 (Cat# ab134113, 1:2,000; Abcam, USA), and rabbit anti-GCase (Cat# ab128879, 1:1,000; Abcam, USA). After three 10-min washes with TBST, membranes were incubated with HRP-conjugated anti-rabbit IgG (Cat# SE134, 1:5,000; Solarbio, China) or anti-mouse IgG (Cat# SE131, 1:5,000; Solarbio, China) secondary antibodies for 2 h at room temperature. Protein bands were visualized using Immobilon Western Chemiluminescent HRP Substrate (Cat# WBKLS0500, Merck Millipore, USA) on a ChemiDoc™ Gel Imaging System using Image Lab software version 6.0 (Bio-Rad, USA). Band intensities were quantified using ImageJ software version 1.53k (National Institutes of Health, USA).

## Cycloheximide treatment

To ensure that the viral signals detected originated solely from incoming virions and not from de novo protein synthesis, the translation inhibitor cycloheximide (CHX, Cat# HY-12320, MedChem-Express, USA) was employed in all entry-related assays (including binding, internalization, endosomal trafficking, fusion, and nuclear import). Cells were pretreated with 50 μg/mL CHX for 1 h prior to infection, and CHX was maintained throughout the infection period until fixation or lysis for analysis.

## Time-of-addition assay

To delineate the effective phase of drug action, time-of-addition assays were performed. A549 cells were pretreated with 50 μg/mL CHX for 1 h and then infected with PR8 at an MOI of 0.1. CHX was maintained throughout the experiment. AmB (1 μg/mL) or CBE (100 μM) was added at specified times relative to infection: −1 h (pre-treatment), 0 h (co-treatment), or at +1, +2, and +4 h (post-treatment). Cells were harvested and lysed at 6 hpi, and the lysates were used for RT-qPCR analysis of viral *nucleoprotein (NP)* RNA levels.

## Viral binding, internalization, and endosomal trafficking assays

A549 cells were incubated with PR8 at 4 °C for 1 h to allow viral attachment while preventing internalization. Unbound virions were removed by washing the cells three times with ice-cold PBS. The binding efficiency was quantified using two parallel approaches: RT-qPCR analysis of viral *NP* RNA levels and immunofluorescence staining under non-permeabilized conditions. For the latter, cells were co-stained with mouse anti-IAV NP antibody (Cat# ab20343, 1:100; Abcam, USA) and biotinylated *Sambucus nigra* lectin (Cat# B-1305-2, 1:200; Vector, Germany) to label sialic acid residues on the plasma membrane. As a negative control, parallel cell samples were pretreated with 50 mU/mL neuraminidase (Cat# 11080725001, Merck, Germany) for 1 h at 37 °C before virus binding.

After viral binding, cells were incubated at 37 °C to allow internalization, followed by NP immunostaining to evaluate viral uptake. In parallel, to assess endosomal trafficking, virus-loaded cells were co-stained with anti-IAV NP (Cat# ab20343, 1:100; Abcam, USA) and either rabbit anti-EEA1 (Cat# 3288, 1:100; Cell Signaling Technology, USA) or rabbit anti-RAB7 (Cat# ab137029, 1:100; Abcam, USA). Colocalization was quantified using the JACoP plugin in ImageJ software version 1.53k (National Institutes of Health, USA), applying the Manders' M2 coefficient (channel A: viral NP; channel B: organelle marker).

## Virus fusion assay

PR8 was concentrated to 100 μg/mL and labeled with 0.4 μM R18 (Cat# O246) and 0.2 μM SP-DiOC18 (Cat# D7778, Thermo Fisher Scientific, USA) following established protocols[27]. The virus-dye mixture was vortexed vigorously for 1 h at room temperature and then filtered through a 0.22 μm filter (Merck Millipore, USA). 10 μL of the labeled virions was incubated with A549 cells seeded in 24-well plates on ice for 1 h to allow binding, followed by three washes with ice-cold PBS to remove unbound particles. Internalization and fusion were initiated by transferring the plates to 37 °C for specified durations. The cells were subsequently fixed with 4% PFA (Beyotime, China) for 10 min, permeabilized with 0.2% Triton X-100 (Beyotime, China) for 10 min, and counterstained with DAPI (Yeasen, China). Imaging was performed using an LSM 800 Confocal Laser Scanning Microscope running ZEN 3.4 Fine Software (Zeiss, Germany) equipped with a 63× oil immersion objective. Viral fusion events were identified by an increased intensity of green fluorescence, indicating the release of self-quenching SP-DiOC18 and the dissolution of FRET. Fusion efficiency was quantified by calculating the Manders' M2 coefficient using ImageJ software version 1.53k (National Institutes of Health, USA), where the SP-DiOC18 signal was defined as channel A and the R18 signal as channel B.

## Plasma membrane bypass infection assay

The protocol was adapted from established methods[48–50]. Briefly, A549 cells were pretreated with 50 μg/mL CHX for 1 h and then infected with

PR8 (MOI = 0.1) in ice-cold medium for 1 h. After washing with ice-cold PBS to remove unbound virus, plasma membrane fusion was triggered by incubating cells for 2 min at 37 °C with prewarmed acidic fusion medium (DMEM containing 0.2% BSA, pH 5.0, adjusted with citrate buffer). Cells were immediately washed with ice-cold PBS and incubated for 5 h at 37 °C in infection medium (DMEM with 0.2% BSA) containing 10 mM NH$_4$Cl and CHX. Finally, cells were harvested for RT-qPCR quantification of viral *NP* RNA.

## Acidic organelle pH measurement

Acidic organelle pH was measured ratiometrically using LysoSensor™ Yellow/Blue DND-160 (Cat# L7545, Thermo Fisher Scientific, USA), following a protocol adapted from previous reports[51]. For calibration, A549 cells were incubated with 1 µM LysoSensor™ in complete medium for 5 min at 37 °C, followed by two washes with PBS. The cells were then treated for 10 min with 10 µM monensin (Cat# S2324, Selleck, USA) in 25 mM MES (Aladdin, China) calibration buffers (pH 3.5–6.0) containing 25 mM NaCl (Merck Millipore, USA), 125 mM KCl (Aladdin, China), and 25 mM HEPES (Aladdin, China) to equilibrate intracellular compartments with the extracellular pH. Fluorescence was measured using a Spark® multimode microplate reader running SparkControl™ software version 2.2 (Tecan, Switzerland), with excitation at 329 nm and 384 nm, and emission recorded at 440 nm and 540 nm, respectively. A pH calibration curve was then generated by plotting the 440/540 nm emission ratio against the buffer pH. For experimental measurements, A549 cells left untreated or pretreated with AmB or Baf-A1 for 2 h were loaded with 1 µM LysoSensor™ under identical conditions, and the 440/540 nm emission ratios were converted to pH values using the established calibration curve.

## Isolation of subcellular fractions

Subcellular fractionation was performed according to established protocols[52] with some modifications. Briefly, A549 cells were grown to 90% confluence in 15-cm culture dishes, washed three times with ice-cold PBS, and harvested in 5 mL PBS supplemented with a protease inhibitor cocktail (Beyotime, China) using cell scrapers. The cell suspension was then centrifuged at 200 × *g* for 5 min at 4 °C. The resulting pellets were washed with 3 volumes of homogenization buffer (HB, 250 mM sucrose and 3 mM imidazole in double-distilled water, pH 7.4) containing phosphatase and protease inhibitors (Beyotime, China), followed by centrifugation at 1300 × *g* for 10 min at 4 °C. The washed pellets were gently resuspended in 3 volumes of HB+ (HB supplemented with 1 mM EDTA and 0.03 mM CHX) and homogenized by passing the suspension through a 22-gauge needle ten times. Homogenization efficiency was verified by microscopic quantification, with a ratio of 70–80% clear nuclei to total cells. Subsequently, 700 µL of HB+ was added per 1 mL homogenate, and the nuclear pellets were separated from the post-nuclear supernatant (PNS), which contains cytosolic components, by centrifugation at 2000 × *g* for 10 min at 4 °C.

For the isolation of endosomal fractions, the PNS was mixed with a 62% sucrose solution at a ratio of 1:1.2 to achieve a final sucrose concentration of 40.6%. The resulting mixture was loaded into SW41Ti centrifuge tubes (Beckman Coulter, USA) and sequentially overlaid with 1.5 volumes of 35% sucrose solution, followed by 1 volume of 25% sucrose solution, and finally topped with HB+ to fill the tube. Ultracentrifugation was performed at 210,000 × *g* for 1.5 h at 4 °C using an Optima XPN-100 ultracentrifuge (Beckman Coulter, USA) equipped with an SW41Ti swinging-bucket rotor, resulting in the formation of two distinct endosomal bands. The upper milky band at the interface between HB+ and 25% sucrose corresponded to late endosomes, whereas the lower band at the 25%/35% sucrose interface contained early endosomes. Fraction collection was performed by careful aspiration from the respective interfaces. Prior to downstream analyses, protein concentrations of all isolated fractions were determined

using the Pierce™ BCA Protein Assay Kit (Cat# 23225, Thermo Fisher Scientific, USA) for standardization.

## Endocytic pathway assays

To evaluate the effects of AmB on multiple endocytic routes, internalization of pathway-specific probes was measured: Alexa Fluor 488-conjugated transferrin (TF-488; Cat# T13342, Thermo Fisher Scientific, USA) for clathrin-mediated endocytosis, tetramethylrhodamine-labeled 70-kDa dextran (TMR-Dextran; Cat# D1818, Thermo Fisher Scientific, USA) for macropinocytosis, and Alexa Fluor 555-conjugated cholera toxin subunit B (CTB-555; Cat# C22843, Thermo Fisher Scientific, USA) for caveolin-dependent uptake. Corresponding pharmacological inhibitors—Pitstop 2 (20 µM; Cat# HY-115604, MedChemExpress, USA), EIPA (50 µM; Cat# HY-101840, MedChemExpress, USA), and methyl-β-cyclodextrin (MβCD, 5 mM; Cat# HY-101461, MedChemExpress, USA)—served as positive controls and were applied for the indicated times prior to and during probe uptake. All assays were performed under light-protected conditions. Cells were seeded on coverslips for confocal microscopy or in 6-well plates for flow cytometry.

Clathrin-mediated endocytosis was assessed by transferrin internalization. Cells were pre-chilled on ice for 10 min and washed. Internalization was initiated by incubating with 25 µg/mL TF-488 at 37 °C for 5 min, followed by an acidic wash (0.2 M glycine, pH 3) to remove surface-bound probe. For fluorescence microscopy, after fixation, permeabilization, and DAPI counterstaining, cells were imaged using a LSM 800 confocal laser scanning microscope running ZEN 3.4 Fine software (Zeiss, Germany). Macropinocytosis was measured using TMR-Dextran uptake. Cells were incubated with 1 mg/mL TMR-dextran at 37 °C for 40 min, followed by extensive washing with ice-cold PBS. Caveolin-dependent uptake was evaluated by CTB-555 internalization. Cells were pre-cooled, incubated with 10 µg/mL CTB-555 on ice for 10 min to allow binding, then shifted to 37 °C for 30 min to trigger internalization.

For flow cytometry, cells were trypsinized and resuspended in PBS containing 2% FBS (Gibco, USA). To exclude dead cells, samples were stained with the LIVE/DEAD™ Fixable Near-IR (876) Dead Cell Stain Kit (Cat# L34982, Thermo Fisher Scientific, USA) according to the manufacturer's instructions. Samples were then analyzed on a CytoFLEX LX Flow Cytometer using CytExpert software version 2.5.0.77 (Beckman Coulter, USA). At least 10,000 events were acquired per sample with appropriate laser/detector settings for each fluorophore. Vehicle-treated controls were included in all experiments.

## Immunofluorescence staining

Cellular localization was examined using standardized immunofluorescence protocols. Cells grown on coverslips were fixed in 4% PFA (Beyotime, China) for 10 min. Except for the virus-binding assay, cells were permeabilized with 0.2% Triton X-100 (Beyotime, China) for 10 min. All samples were then blocked with Immunol Staining Blocking Buffer (Beyotime, China) for 1 h. Fixation, permeabilization (where applicable), and blocking were all performed at room temperature.

Primary antibodies—mouse anti-IAV NP (Cat# ab20343, 1:100; Abcam, USA), biotinylated *Sambucus nigra* lectin (Cat# B-1305-2, 1:200; Vector, Germany), rabbit anti-EEA1 (Cat# 3288, 1:100; Cell Signaling Technology, USA), and rabbit anti-RAB7 (Cat# ab137029, 1:100; Abcam, USA)—were applied in a humidified chamber at 4 °C overnight. After three 5-min washes with TBST, appropriate secondary antibodies —including Alexa Fluor 488- (Cat# A32766, 1:1,000) and 555-conjugated (Cat# A31570, 1:1,000) donkey anti-mouse IgG, Alexa Fluor 555-conjugated donkey anti-rabbit IgG (Cat# A32794, 1:1,000; Thermo Fisher Scientific, USA), and Alexa Fluor 488-conjugated Streptavidin (Cat# 405235, 1:400; BioLegend, USA)—were incubated for 2 h at room temperature in the dark. Primary and secondary reagents were selected and applied according to the requirements of each specific assay.

Following this incubation, cells were washed three times with TBST before nuclear counterstaining with DAPI (Yeasen, China). Images were acquired using an LSM 800 Confocal Laser Scanning Microscope running ZEN 3.4 Fine Software (Zeiss, Germany).

## Synthesis and characterization of the AmB-biotin conjugate

A biotinylated derivative of AmB (AmB-biotin) was synthesized and characterized to facilitate further analysis. All synthesis and analytical measurements were conducted by Pharmaron (USA). Solvents were purchased from Sigma-Aldrich (USA) and used without further purification. Reactions were performed in anhydrous solvents under a nitrogen atmosphere. Proton NMR spectra were recorded using a Bruker Plus 400 NMR Spectrometer (Germany), with deuterated solvents containing 0.03–0.05% v/v tetramethylsilane as an internal reference ($\delta$ 0.00 for both $^1$H and $^{13}$C). Preparative reverse-phase high-performance liquid chromatography (HPLC) was conducted on a Varian HPLC system (USA) equipped with an XBridge Prep C18 OBD Column (5 μm, 19 × 150 mm, Waters, USA) under reverse-phase conditions using acetonitrile/water containing 0.1% ammonium hydrogen carbonate or formic acid.

In a representative procedure, (1R,3S,5R,6R,9R,11R,15S,16S,17R, 18S,19E,21E,23E,25E,27E,29E,31E,33R,35S,36R,37S)-33-{[(2R,3S,4S,5S, 6R)-4-amino-3,5-dihydroxy-6-methyloxan-2-yl]oxy}-1,3,5,6,9,11,17,37-octahydroxy-15,16,18-trimethyl-13-oxo-14,39-dioxabicyclo[33.3.1]non-atriaconta-19,21,23,25,27,29,31-heptaene-36-carboxylic acid (6.09 g, 6.591 mmol, 0.5 eq.) was dissolved in 100 mL DMF with Et$_3$N (8.00 g, 79.092 mmol, 6 eq.) in a 500 mL three-necked round-bottom flask maintained at 0 °C. To this solution, 2,5-dioxopyrrolidin-1-yl 5-[(3aS,4S,6aR)-2-oxo-hexahydrothieno[3,4-d]imidazol-4-yl]pentanoate (4.5 g, 13.182 mmol, 1 eq.) was added, and the reaction mixture was stirred at room temperature overnight under a nitrogen atmosphere. The precipitated solids were collected by filtration and washed three times with 100 mL portions of ethanol. The crude product was purified by preparative HPLC using an Xselect CSH Prep C18 Column (5 μm, 30 × 150 mm) with water (0.1% formic acid) as mobile phase A and acetonitrile as mobile phase B, at a flow rate of 60 mL/min. A gradient from 40% to 70% B over 10 min was used, monitored at 254 nm/220 nm, with a retention time of 8.5 min.

## Pull-down and LC–MS/MS

To identify potential late endosome-associated targets of AmB, pull-down experiments combined with silver staining and LC–MS/MS analysis were performed. A549 cells were cultured in 15-cm culture dishes to 90% confluency and infected with PR8 at an MOI of 1 for 4 h. Late endosomes were subsequently isolated as described in the Isolation of subcellular fractions section. Protein concentration was determined using the Pierce™ BCA Protein Assay Kit (Cat# 23225, Thermo Fisher Scientific, USA). Streptavidin magnetic beads (Cat# HY-K0208, MedChemExpress, USA) were employed to capture biotinylated AmB–protein complexes via magnetic separation. Three experimental groups were analyzed: biotin control, AmB-biotin, and AmB-biotin with excess free AmB. Beads were incubated with 50 nmol of biotin (control group) or AmB-biotin (experimental group) for 1 h at room temperature. After washing three times, the beads were incubated with 50 μg of the late endosomal fraction for 1 h, either with 500 nmol of free AmB (competition group) or without (control/AmB-biotin groups). Following another wash, bound proteins were eluted by boiling in SDS-PAGE loading buffer, resolved on 4–20% gradient SDS-PAGE gels (ACE Biotechnology, China), and visualized via silver staining (Beyotime, China). Protein bands that were present in the AmB–biotin group, absent in the biotin control, and exhibited reduced intensity in the competition group were excised for further analysis. The excised gel slices were subjected to trypsin digestion and the resulting peptides were analyzed by LC–MS/MS, which was performed by Bioprofile (China) using an LTQ Velos Pro mass spectrometer (Thermo Fisher Scientific, USA). Peptide samples were filtered through 0.22 μm membranes and loaded onto a Captrap Peptide column (Thermo Fisher Scientific, USA) at a flow rate of 20 μL/min. Subsequent separation was performed on a C18AQ reverse-phase column (Thermo Fisher Scientific, USA) with dimensions of 100 μm inner diameter by 15 cm, maintained at 35 °C using an electrospray voltage of 1.8 kV.

## Surface plasmon resonance (SPR)

The interaction between GCase and AmB was analyzed by MedChemExpress (USA) using a Biacore T200 system (Cytiva, USA). Recombinant human GCase protein (Cat# HY-P75786, MedChemExpress, USA) was immobilized on a CM5 sensor chip (Cytiva, USA) via standard amine coupling. First, the chip surface was activated with a mixture of N-ethyl-N'-(3-dimethylaminopropyl) carbodiimide (EDC, Cytiva, USA) and N-hydroxysuccinimide (NHS, Cytiva, USA) at a flow rate of 10 μL/min. GCase, diluted to 50 μg/mL in sodium acetate buffer, was then injected over the activated surface at the same flow rate. Unreacted sites were subsequently blocked with ethanolamine, also delivered at 10 μL/min. AmB (MedChemExpress, USA) was prepared in a series of concentrations (0, 3.125, 6.25, 12.5, 25, 50, and 100 μM) and injected in ascending order at 30 μL/min, with each cycle including an association phase of 150 s followed by a dissociation phase. Between each cycle, the sensor surface was regenerated using 10 mM glycine-HCl (pH 2.0) for 5 min to remove any bound analytes. Real-time binding responses were recorded in resonance units and corrected for nonspecific binding using a reference channel and appropriate buffer controls. Kinetic parameters were derived by globally fitting the data to a 1:1 Langmuir binding model using Biacore T200 Evaluation Software and Biacore Insight Software (Cytiva, USA).

## GCase activity assay

The enzymatic activity of GCase was determined using the Glucosylceramidase Activity Assay Kit (Cat# ab273339, Abcam, USA) following the manufacturer's instructions. Confluent cells in 10-cm culture dishes were collected and lysed with 200 μL of ice-cold Assay Buffer XXV for 10 min, and the lysates were centrifuged at 12,000 × $g$ for 10 min at 4 °C to remove cellular debris. For each assay, 20 μL each of the clarified supernatant, Assay Buffer XXV, and a 1:20 dilution of the fluorescent substrate were combined in 96-well flat-bottom white microplates (Thermo Fisher Scientific, USA) and incubated at 37 °C for 30 min. The reaction was halted by adding 100 μL of Stop Solution, and fluorescence was immediately measured on a Spark® multimode microplate reader running SparkControl™ software version 2.2 (Tecan, Switzerland) with excitation at 360 nm and emission at 445 nm. Total protein concentration was measured in parallel using the Pierce™ BCA Protein Assay Kit (Cat# 23225, Thermo Fisher Scientific, USA), and the fluorescence values were normalized to the total protein concentration. A standard curve was generated from serially diluted reference standards by plotting fluorescence intensity against the amount of released 4-methylumbelliferone. One unit of GCase activity is defined as the amount of enzyme that produces 1.0 μmol of 4-methylumbelliferone per minute at pH 4.5 and 37 °C, according to the kit specifications.

## Lipidomics analyses

Lipid extraction from late endosomes was performed using a modified Bligh and Dyer's method[53]. Briefly, isolated late endosomes were homogenized in 750 μL of chloroform: methanol: MilliQ H$_2$O (3:6:1, v/v/v) and incubated at 1500 rpm for 1 h at 4 °C. Phase separation was induced by adding 350 μL of deionized water and 250 μL of chloroform, followed by centrifugation at 2000 × $g$ for 10 min. The lower organic phase containing lipids was transferred to a clean tube, and residual lipids in the aqueous phase were re-extracted with 450 μL of chloroform. The combined lipid extracts were dried using a SpeedVac concentrator (Thermo Fisher Scientific, USA) under OH mode and

stored at −80 °C until analysis, while the upper aqueous phase and pellet were dried separately under $H_2O$ mode. In parallel, total protein concentration was determined using the Pierce™ BCA Protein Assay Kit (Cat# 23225, Thermo Fisher Scientific, USA).

Lipidomic profiling was conducted at LipidALL Technologies (China) using an ExionLC-AD system (Sciex, USA) coupled to a QTRAP 6500 PLUS mass spectrometer (Sciex, USA). Polar lipid separation was achieved by normal-phase HPLC on a TUP-HB silica column (150 × 2.1 mm, 3 μm, TUP, China) using mobile phase A (chloroform: methanol: ammonium hydroxide, 89.5:10:0.5) and mobile phase B (chloroform: methanol: ammonium hydroxide:water, 55:39:0.5:5.5). Targeted lipid quantification was performed using multiple reaction monitoring (MRM) transitions, and internal standards, including $d_9$-PC32:0(16:0/16:0), $d_9$-PC36:1p(18:0p/18:1), Cer d18:1/15:0-$d_7$, $d_9$-SM d18:1/18:1, C8-GluCer, C8-GalCer, $d_3$-LacCer d18:1/16:0, Gb3 d18:1/17:0, $d_7$-LPC18:1, d17:1 Sph, and d17:1 S1P (Avanti Polar Lipids, USA), which were used for absolute quantification. Lipid concentrations were normalized to protein content, with results expressed as μmol/g protein.

## Statistical analysis

For the animal, cell, and molecular biology studies, data are presented as the mean ± SEM. Sample sizes (n) represent the number of independent biological replicates and are explicitly stated in the corresponding figure legends. Statistical analyses were performed using GraphPad Prism 9.5.1 (USA). Comparisons between two groups were evaluated using two-tailed unpaired Student's t-tests. For comparisons among more than two groups, one-way analysis of variance (ANOVA) was performed, followed by either Tukey's or Dunnett's multiple comparisons test, as appropriate. When experiments involved multiple groups with two independent variables, two-way ANOVA was conducted, followed by Tukey's, Dunnett's, or Bonferroni's multiple comparisons test, as indicated in the figure legends. Statistical significance was defined as $P < 0.05$.

For the retrospective clinical data analysis, categorical variables are presented as n (%), and continuous variables as median (interquartile range). The Mann–Whitney U test was used for continuous variables, and the $\chi^2$ test or Fisher's exact test for categorical variables, as appropriate. To control for potential confounders, propensity score matching (PSM) was performed in a 1:1 ratio using covariates including demographics, vital signs, comorbidities, treatments, and clinical outcomes. Group characteristics were compared before and after PSM using the appropriate tests mentioned above. Additionally, multivariable logistic regression analysis was conducted on the total study population. Candidate variables, including age, sex, major comorbidities, and the use of immunosuppressive medications, were evaluated. Variables with $P < 0.05$ in the univariable analysis were included in the multivariable model using a forward stepwise selection procedure, with AmB treatment as the primary exposure. The primary outcome was defined as the occurrence of laboratory-confirmed respiratory viral infection following the initiation of antifungal therapy. Statistical analyses were performed using R 4.3.3 (Austria), and statistical significance was defined as $P < 0.05$.

## Ethics statement

All animal experiments were conducted in strict accordance with the Guide for the Care and Use of Medical Laboratory Animals (Ministry of Health, People's Republic of China) and received approval from two independent ethics committees. Specifically, the protocols for the murine experiments were evaluated and approved by the Animal Ethics Committee of the Institute of Biophysics, Chinese Academy of Sciences (Approval number ABSL-2-2023019), whereas the procedures for hamster experiments were reviewed and sanctioned by the Institutional Animal Care and Use Committee of the Institute of Laboratory Animal Science, Peking Union Medical College & Chinese Academy of Medical Sciences (Approval number XZG23001). IAV challenges were

carried out in animal biosafety level 2 (ABSL-2) containment, while SARS-CoV-2 infections were performed in animal biosafety level 3 (ABSL-3) facilities in accordance with WHO interim guidance.

In addition, the retrospective clinical cohort study was performed in line with the principles of the Ethics Committee of China-Japan Friendship Hospital (Approval number 2022KY-052) in accordance with the Declaration of Helsinki. The requirement for informed consent was waived by the Ethics Committee due to the retrospective nature of the study, and no participant compensation was provided.

## Reporting summary

Further information on research design is available in the Nature Portfolio Reporting Summary linked to this article.

## Data availability

The mass spectrometry proteomics data generated in this study have been deposited in the ProteomeXchange Consortium under accession codes PXD064601 and PXD064637. The targeted lipidomics data are provided in Supplementary Data 1. Raw lipidomics instrument files are unavailable due to third-party intellectual property restrictions associated with the service provider (LipidALL Technologies), but the provided quantitative dataset is sufficient to interpret and validate the reported findings. Patient clinical data are not publicly available to strictly comply with data privacy laws and ethical regulations protecting patient confidentiality. However, de-identified clinical data will be made available to researchers upon reasonable request. Access requests should be submitted to the corresponding author and must include a research proposal and a signed Data Use Agreement. The corresponding author will review requests in accordance with the ethics committee's requirements and respond within 4 weeks. All other data supporting the findings of this study are available within the article, its Supplementary Information, or the Source Data file. Source data are provided with this paper.

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

## Acknowledgements

We thank Dr. Linqi Zhang and Dr. Xuanling Shi from Tsinghua University for providing the plasmids CMV/R-HA and CMV/R-N1NA; Dr. Ning Jiao and Dr. Xiaodong Dou from Peking University for their discussions on the experimental framework; Dr. Hui Wang from Peking University People's Hospital for her comments on the manuscript; Dr. Xiaoying Gu and Dr. Rongling Zhang from China-Japan Friendship Hospital for their suggestions on statistical analysis; Dr. Fei Zhu from Xiangya Hospital for assistance with chemical structures; and Wenqing He from China Medical University for help with figure preparation and data verification. The work was supported by the National Natural Science Foundation of China (Nos. 82241056 and 82530002 to B.C., Nos. 82170015 and 82570008 to Z.W., No. 824B2001 to W.Z.), the Chinese Academy of Medical Sciences (CAMS) Innovation Fund for Medical Sciences (Nos. CIFMS 2021-I2M-1-048 and 2024-I2M-ZD-011 to B.C.), the Excellence & Innovation Initiative of China-Japan Friendship Hospital (No. ZRZC2025-KCB03 to Z.W.), the New Cornerstone Science Foundation (to B.C.), and the Fundamental Research Funds for the Central Universities (No. APL24200210010302003 to W.Z.).

## Author contributions

B.C., Z.W., and P.W. conceived and supervised the study. Z.W. and M.L. designed the experimental framework. D.H., W.Z., Y.H., Y.Y., W.W., and Y.L. carried out the in vitro cellular experiments. H.L., Q.F., X.L., and Yun Z. performed the molecular biology assays. Z.X., W.T., and Ying Z. executed the animal infection studies. D.H., J.Z., X.Z., and D.P. collected and analyzed the clinical coinfection data. D.H. and W.Z. conducted data analysis and drafted the manuscript. B.C., Z.W., and W.Z. obtained funding for the study. All authors contributed to the manuscript preparation and approved the final version.

## Competing interests

The authors declare no competing interests.

## Additional information

**Di He** [1,2,13], **Wenting Zuo**[2,3,13], **Zhiguang Xiang**[4,13], **Jiankang Zhao**[2,13], **Wei Tong**[4], **Hongyan Li**[5], **Qing Fang**[5], **Xin Li**[5], **Yun Zhang**[5], **Ying Zheng**[1,2], **Xianxia Zhuo**[1,2], **Danni Pu**[2,3], **Yijiao Huang**[2,6], **Yingying Yuan**[2,7], **Weiyang Wang**[2,3], **Yameng Lu**[2], **Min Luo** [8] ✉, **Peigang Wang** [9,10] ✉, **Zai Wang** [2,5,7] ✉ & **Bin Cao** [1,2,3,11,12] ✉

[1]Department of Pulmonary and Critical Care Medicine, China-Japan Friendship Hospital, Capital Medical University, Beijing, China. [2]National Center for Respiratory Medicine; State Key Laboratory of Respiratory Health and Multimorbidity; National Clinical Research Center for Respiratory Diseases; Institute of Respiratory Medicine, Chinese Academy of Medical Sciences & Peking Union Medical College; Department of Pulmonary and Critical Care Medicine, Center of Respiratory Medicine, China-Japan Friendship Hospital, Beijing, China. [3]Peking Union Medical College, Chinese Academy of Medical Sciences, Beijing, China. [4]National Human Diseases Animal Model Resource Center, National Center of Technology Innovation for animal model, NHC Key Laboratory of Comparative Medicine, Institute of Laboratory Animal Science, CAMS & PUMC, Beijing, China. [5]Institute of Clinical Medical Sciences, China-Japan Friendship Hospital, Beijing, China. [6]School of Basic Medical Sciences, Tsinghua University, Beijing, China. [7]Peking University China-Japan Friendship School of Clinical Medicine, Beijing, China. [8]Institute of Pediatrics of Children's Hospital of Fudan University, the Shanghai Key Laboratory of Medical Epigenetics, the International Co-laboratory of Medical Epigenetics and Metabolism, Ministry of Science and Technology, Institutes of Biomedical Sciences, Fudan University, Shanghai, China. [9]Department of Microbiology, School of Basic Medical Sciences, Capital Medical University, Beijing, China. [10]Laboratory for Clinical Medicine, Capital Medical University, Beijing, China. [11]Tsinghua University-Peking University Joint Center for Life Sciences, Beijing, China. [12]New Cornerstone Science Laboratory, Department of Pulmonary and Critical Care Medicine, Center of Respiratory Medicine, China-Japan Friendship Hospital, Beijing, China. [13]These authors contributed equally: Di He, Wenting Zuo, Zhiguang Xiang, Jiankang Zhao. ✉e-mail: luo_min@fudan.edu.cn; pgwang@ccmu.edu.cn; wzai_163pass@163.com; caobin_ben@163.com

