## [Transparent Peer Review file · Nature Communications]

Amphotericin B Promotes Respiratory Viral Entry by Enhancing Late Endosomal Maturation and Fusion via Glucocerebrosidase-Mediated Ceramide Remodeling

Corresponding Author: Professor Bin Cao

Version 0:

Reviewer comments:

Reviewer #1

(Remarks to the Author)

Fungal coinfections in influenza and COVID-19 patients are associated with worse outcome and increased case-fatality rates. This study examines the effect of anti-fungal treatment with AmB on IAV and SARS-CoV-2 pathogenesis. In two rodent infection models, mice for IAV and Syrian golden hamsters for SARS-CoV-2, a positive correlation between AmB administration and enhanced clinical signs and lung histopathology was noted. Mechanistic characterization established a link between AmB administration and viral membrane fusion efficiency and identified AmB targeting of GCCase in late endosomes, which upregulates ceramide production and modulates endosome protein composition and maturation. The study is very comprehensive by design, the topic is highly timely, and clinical relevance of the findings is high. Polypharmacological drugs such as AmB are frequently associated with unexpected effects, and the authors make a commendable effort to delineate the mechanistic underpinning for seemingly contradictory findings in previous studies concerning the impact of AmB on diverse viral infections. The data presented here suggest that treatment of fungal coinfections of primary respiratory virus infections with AmB may indeed be counterproductive, calling for further clinical evaluation. Several specific points require clarification, however, for full support of the author's conclusions.

Specific points:

- 1) Do the relative viral RNA/protein differences shown in Fig. 1C and 1J translate into actual differences in infectious virus load? Considering that AmB has also been associated with reduced maturation of viral glycoproteins and altered integrity of viral particles in the past, please determine actual infectious virus titers in lung tissue samples.
- 2) Presumably, the hospitalized patients that received AmB had more severe fungal co-infections compared to those that did not receive AmB. Please clarify as much as the records allow severity of the fungal coinfections in the different patient groups. If there is a positive correlation between administration of AmB and severity of fungal infection, poorer clinical outcome of this group must be anticipated independent of potential exacerbated viral pathogenesis by AmB. Please outline your strategy for separation of AmB vs fungal co-infection-induced differences in outcome.
- 3) Very little particle binding (Fig. 2J) and very few internalization events (Fig. 2L) can be seen. It is unclear how these images correlate to the 5-10% positive cells claimed in the quantitations. Can you show fields of view that are more supportive of the quantitative data provided?
- 4) What are we looking at in fig. 5C? A meaningful legend explaining the color-intensity coding would be helpful.
- 5) Lines 74-75: the same antiviral mechanism of AmB has been proposed for HIV (i.e. PMID: 7983757). Worthwhile to add here.
- 6) Histopathology slides (Fig 1E and K) appear to be overstained.

Reviewer #2

(Remarks to the Author)

The study by He et al. investigates the effects of amphotericin B (AmB), a widely used antifungal drug, on respiratory viral infections, specifically influenza A virus (IAV) and SARS-CoV-2. While AmB plays a critical role in treating invasive fungal infections, particularly in patients co-infected with viruses, the authors present evidence that AmB exacerbates viral infections by enhancing viral entry. Mechanistically, the study reveals that AmB binds to and activates glucocerebrosidase (GCCase), a lysosomal enzyme. This activation increases ceramide levels in late endosomes, promoting membrane fusion

between the virus and the host cell, thereby facilitating viral genome release into the cytosol and enhancing replication. Notably, this effect is specific to late-penetrating viruses like IAV and SARS-CoV-2, and does not extend to early-penetrating viruses such as VSV, which fuse at early endosomes. The authors demonstrate that AmB enhances the entry of IAV and SARS-CoV-2 pseudoviruses across various cell types, independently of the antiviral host protein IFITM3. This enhancement depends on GCase activity, as both pharmacological inhibition and genetic knockout of GBA1 suppress AmB-mediated viral entry. In vivo studies further support these findings: AmB treatment in IAV-infected mice and SARS-CoV-2-infected hamsters led to greater weight loss, higher viral loads, and more severe lung pathology compared to untreated controls. Additionally, retrospective clinical data from patients co-infected with influenza/SARS-CoV-2 and *Aspergillus* showed that those receiving systemic AmB had significantly longer ICU stays and mechanical ventilation durations, although mortality rates were not significantly different. At the molecular level, proteomic analysis revealed that AmB treatment upregulates RAB7 in a GCase-dependent manner, contributing to enhanced viral fusion and entry. The authors propose a model in which AmB-induced GCase activation and ceramide accumulation remodel late endosomal membranes to favor viral fusion. These findings challenge current assumptions regarding the safety of AmB in virally co-infected patients and suggest that antifungal treatment regimens may need to be reassessed during viral outbreaks. Nevertheless, several major and a few minor issues remain to be addressed in this study.

Major comments:

1. Although AmB treatment did not affect viral binding to the plasma membrane, as shown in Fig. 2I, an increase in NP levels was observed in AmB-treated cells at 2, 6, and 12 hpi (Fig. 2G). Notably, treatment with 1 $\mu\text{g/ml}$ AmB significantly upregulated NP expression at 6 hpi, comparable to the effect seen with 0.3 $\mu\text{g/ml}$. However, at 2 hpi, 1 $\mu\text{g/ml}$ AmB appeared to have no effect, and the increase at 12 hpi was modest. This discrepancy in the temporal response should be addressed by the authors. Additionally, the statistical significance of the 2 hpi data in Fig. 2G should be shown. The findings would be further strengthened by including known inhibitors of IAV entry, such as bafilomycin A1 or ammonium chloride, as positive controls.
2. Since IAV entry in A549 cells occurs rapidly, with most viral particles reaching the late endosomes within an hour, earlier time points should be examined. Assessing later time points such as 6 h or 12 h may not accurately reflect entry-related events.
3. In viral entry assays, cycloheximide is commonly used to block host translation, ensuring that detected NP (or other viral components) originates from incoming virions rather than newly synthesized proteins. However, it appears that cycloheximide was not used in the current experiments. To accurately assess viral entry, all relevant assays should be repeated with cycloheximide treatment.
4. The images in Fig. 2J and 2K do not convincingly demonstrate viral binding. For a proper IAV binding assay, surface-bound viruses should be visualized at higher magnification under unpermeabilized conditions using a membrane or cell surface marker. The NP signals appear discrete and patchy, which is unexpected at an MOI of 0.1, where viral particles are typically distributed across most cells, not limited to a few "NP-positive" cells. Quantification should be based on relative fluorescence intensity of viral proteins at the plasma membrane, ideally compared to controls such as no-virus or neuraminidase-treated cells.
5. Influenza A virus (IAV) is known to enter cells via both clathrin-mediated and clathrin-independent endocytic pathways. However, the effect of AmB has been assessed only in the context of clathrin-mediated uptake, using transferrin as a marker. This does not account for other clathrin-independent mechanisms such as macropinocytosis, which IAV is also known to utilize. The authors should evaluate the impact of AmB on additional endocytic pathways, particularly macropinocytosis. Moreover, several experiments lack appropriate controls, which are essential for drawing robust conclusions.
6. The manuscript frequently reports colocalization using Manders' coefficient; however, it does not specify whether M1 or M2 values are being used. This distinction is critical for proper interpretation of the data and should be clearly indicated in all relevant instances throughout the manuscript.
7. The authors claim that at 3 hpi, IAV escapes more rapidly from Rab7-positive late endosomes in AmB-treated cells compared to controls. While Fig. 3B shows increased NP signal in Rab7-negative regions of AmB-treated cells, it is highly unlikely that incoming viruses at this time point would still remain in the cytosol without having entered the nucleus. Based on current understanding, vRNPs released after fusion typically undergo rapid nuclear import, rather than persisting in the cytosol. Moreover, at the current image resolution, it is difficult to accurately assess NP distribution, and the vRNPs appear largely punctate. To better understand whether the vRNPs are still within early endosomes, co-staining of NP with early endosomal markers such as EEA1 is recommended. Additionally, higher-resolution images are essential to reliably determine endosomal localization or escape of viral particles, especially under cycloheximide treatment.
8. To substantiate the claim that AmB treatment accelerates viral escape from late endosomes, a time-course analysis with finer temporal resolution, rather than relying solely on 2 and 3 hpi, should be conducted. Additionally, more rigorous quantitative analysis is necessary, along with appropriate positive controls such as bafilomycinA1 or ammonium chloride. As mentioned above, high-resolution images that clearly depict endosomal structures and viral localization are essential to support this conclusion.
9. In Fig. 3E, since IAV fusion typically occurs between 1 and 2 hpi, the absence of a fusion signal at 2 hpi in control samples requires clarification. Likewise, the near-complete absence of SP-DiOC18 signal at 4 hpi in control cells, as shown in Fig. 3F, is highly unlikely and needs explanation. It is improbable that no SP-DiOC18-positive particles would be detected

in untreated cells. Furthermore, the SP-DiOC18 signal in the images provided appears diffusely distributed throughout the cytosol, rather than localized to endosomal compartments as expected. These observations raise serious concerns about the reliability of this otherwise well-established IAV fusion assay. The experiment should be repeated under cycloheximide treatment, include appropriate positive controls, and be supported by high-magnification images that clearly resolve endosomal localization.

10. The authors state (line 202) that AmB enhances nuclear accumulation of viral NP. However, Fig. 3G shows a strong NP signal in the cytoplasm at 6 hpi. At this time point, such signals are typically attributed to newly synthesized NP, which localizes to both the nucleus and cytoplasm. Therefore, without cycloheximide treatment to inhibit viral protein synthesis, it is not possible to specifically assess vRNP nuclear import during the early entry phase. Additionally, evaluating vRNP nuclear import at late time points such as 9 or 12 hpi is unnecessary and potentially misleading. Instead, a time-course analysis focusing on earlier time points, up to 4 or 5 hpi, would be more appropriate, as nuclear import of vRNPs is known to occur within this window. Appropriate controls such as importazole (importin beta inhibitor) or bafilomycin A1 should be included.

11. To support the claim that AmB promotes IAV escape from late endosomes by enhancing fusion at the endosomal membrane, an acid bypass infection assay should be performed. This approach induces low pH-mediated viral fusion at the plasma membrane, allowing viral genomes to reach the nucleus without endosomal trafficking. If AmB acts specifically at the level of late endosomes, NP expression levels following plasma membrane fusion should be comparable between AmB-treated and control cells.

12. A time-of-addition assay for AmB should be performed, particularly during early time points and in the presence of cycloheximide, to determine whether the observed enhancement in NP expression is truly due to accelerated viral escape from late endosomes.

13. In Fig. S7A, LysoSensor is used to assess endosomal pH; however, critical controls such as bafilomycin A1 or ammonium chloride, which are known to raise endosomal pH, are missing. Additionally, the baseline pH of untreated cells is not indicated and should be included to allow proper interpretation of the results.

14. In Fig. 5A, AmB treatment is applied for 4 h to measure GCCase activity, while in other experiments it is applied for only 1 h prior to IAV infection. Measuring GCCase activity after 1 h of AmB treatment and then at intervals up to 4–6 h would provide a better understanding of the kinetics.

15. The authors convincingly show that GCCase activity in purified endosomes is enhanced upon AmB treatment and propose that this increase promotes viral escape from late endosomes via elevated ceramide levels. While the effect of AmB on GCCase activity appears robust, the current data do not definitively establish whether this impact occurs specifically during viral entry. In the absence of reliable viral entry assays, it remains possible that elevated GCCase activity in other cellular compartments contributes to enhanced viral gene expression at post-entry stages. To address this, experiments should be carefully designed to distinguish whether increased ceramide production directly facilitates viral entry or instead influences downstream steps in the viral life cycle.

16. In addition to testing pseudoviruses in GBA1 KO or KD cells, WT viruses such as PR8 should be tested.

17. In addition to Rab7 and LAMP1, the PCA analysis in Fig. 5 should incorporate other lysosomal hydrolases implicated in late endosomal function and viral entry. Including these markers would provide stronger support for the claim of lysosomal remodeling.

18. In Fig. 6J, quantification of endosomal marker levels from immunoblots is presented without showing the corresponding loading control. This is essential for accurate interpretation.

19. As already mentioned, appropriate controls are missing in many of the experiments.

Minor comments:

1. In line 169, the figure reference should be corrected from Fig. 2G to Fig. 2J.

2. The sequence of subpanels in Fig. 5 should be reorganized for logical flow.

Reviewer #3

(Remarks to the Author)

The authors start by setting up an influenza A mice model and SARS-CoV-2 hamster models which are pre-treated with IV amphotericin B one hour prior to, and then at 24 hr and 72 hr after virus challenge. Compared with controls without amphotericin B treatment, significantly higher weight loss, lung viral load or viral antigen positive cells, histopathological changes of necrotizing pneumonia and serum creatinine levels due to subclinical renal damage were found at day 2 and 5 post-infection. With this finding, they conducted a retrospective clinical study using 2017 to 2019 influenza A patients, and 2022 to 2023 SARS-CoV patients with fungal test positive. 17 patients with Amphotericin B treatment were found to have significantly higher rate of ICU admissions, ICU stay, duration of mechanical ventilation (but not the mortality rate at 28 day) when compared to 163 without amphotericin B treatment. Then they showed that A549 cell line pretreated with Amphotericin

B produced a higher viral load and NP expression after infection by PR8 H1N1 starting at 6 hr postinfection. Cell entry of H5 and SARS-CoV-2 pseudovirus was also augmented in another 3 cell lines by Amphotericin B and Nystatin independent of IFITM3 expression as demonstrated by knockdown and overexpression experiments. Flow cytometry and immunofluorescence analysis showed that Amphotericin B enhances viral entry independent of binding, endocytosis or IFITM3 restriction. Time-resolved co-localization analyses at 3 hr postinfection showed that influenza virions in Amphotericin B treated cells had significantly reduced retention within late endosomes and showed accelerated dispersal toward the nuclear region which suggested an increased escape from late endosomes. A dual fluorescent PR8 virus fusion assay confirmed that Amphotericin B promotes PR8 virus-endosome fusion with concomitant alkalisation of endosomes. Furthermore, amphotericin pretreatment significantly enhanced the entry of HIV/SARS CoV 2 and HIV/MERS CoV pseudoviruses as shown by luciferase assays. Finally, they showed that Amphotericin B-biotin conjugate specifically pull down glucocerebrosidase (encoded by GBA1). Moreover, Amphotericin B treatment significantly enhanced glucocerebrosidase activity in A549 and other cell lines. Chemical inhibition of glucocerebrosidase by conduritol epoxide and CRISPR-Cas9 knockout of GBA1 in 293T cell line reduced enzymatic activity and the amphotericin B mediated enhancement of virus entry in pseudovirus assays of influenza and SARS-CoV-2. Proteomic analysis showed that amphotericin B upregulated late endosomal marker RAB7, caveolin-1 and RAB interacting lysosomal protein on proteomic analysis and Western blot analysis, whereas GBA1 gene deletion downregulated all these markers. The authors concluded that amphotericin B binding triggers a conformational activation of glucocerebrosidase which leads to ceramide accumulation and upregulation of RAB7 and therefore remodels endosomal architecture. These changes promote the fusion of the viral envelope with late endosomal membranes. Therefore, glucocerebrosidase-regulated ceramide synthesis and endosomal maturation can be novel pharmacological targets for modulation of virus infection.

Strength

1. The clinical question is important though mould infection after influenza and SARS-CoV-2 are relatively uncommon unless there are major underlying lung disease, prolonged invasive ventilation and immunosuppression.
2. The experiments are well done, and the paper is well written.

Weakness that must be addressed before the paper can be published:

1. In the usual clinical situation, the patient usually has a long period of influenza A or SARS-CoV-2 disease before the onset of Aspergillus or Mucorales infection, and a significant proportion has been given steroid or biologics immunosuppression before the onset of these mould infections. It would be wrong to give amphotericin B pre-treatment before virus challenge to cell lines and animals.
2. In the retrospective case control study of the clinical patients treated or not treated with amphotericin B, the authors have not provided sufficient clinical data in terms of their infection by which mould species and whether the documentation by culture, antigen or PCR test. They have not provided the data in terms of time of onset of mould infection and the virus infection, and most importantly the amount and duration of immunosuppression by steroid and biologics.
3. Therefore, all the virus infection of cell lines and animal experiments must also be repeated and findings validated by giving amphotericin B after virus challenge. For example, the amphotericin B should be given at least 48 hours after virus challenge which is the usual or normal clinical scenario.

Version 1:

Reviewer comments:

Reviewer #1

(Remarks to the Author)

The authors have made a major effort in their response to previous critiques, adding a substantial amount of new experimental data. My concerns have been addressed.

Reviewer #2

(Remarks to the Author)

I have carefully evaluated the revised manuscript. The authors have now adequately addressed all the points raised by me in my previous review. The revisions have led to a significant improvement in the overall quality of the work. I have no further major concerns and believe the manuscript is now suitable for publication.

Reviewer #3

(Remarks to the Author)

The authors have adequately addressed my concern with additional analysis of clinical data, and repeat cell line and animal experiments with reversal of the order of antifungal treatment.

I have no additional comments.

Reply to Reviewer #1:

Fungal coinfections in influenza and COVID-19 patients are associated with worse outcome and increased case-fatality rates. This study examines the effect of anti-fungal treatment with AmB on IAV and SARS-CoV-2 pathogenesis. In two rodent infection models, mice for IAV and Syrian golden hamsters for SARS-CoV-2, a positive correlation between AmB administration and enhanced clinical signs and lung histopathology was noted. Mechanistic characterization established a link between AmB administration and viral membrane fusion efficiency and identified AmB targeting of GCase in late endosomes, which upregulates ceramide production and modulates endosome protein composition and maturation.

The study is very comprehensive by design, the topic is highly timely, and clinical relevance of the findings is high. Polypharmacological drugs such as AmB are frequently associated with unexpected effects, and the authors make a commendable effort to delineate the mechanistic underpinning for seemingly contradictory findings in previous studies concerning the impact of AmB on diverse viral infections. The data presented here suggest that treatment of fungal coinfections of primary respiratory virus infections with AmB may indeed be counterproductive, calling for further clinical evaluation. Several specific points require clarification, however, for full support of the author's conclusions.

Thank you for your careful review of the manuscript and your high evaluation of our work. We have supplemented the experiments and analysis based on your constructive feedback and made corresponding revisions in the manuscript and supplementary materials.

- 1) Do the relative viral RNA/protein differences shown in Fig. 1C and 1J translate into actual differences in infectious virus load? Considering that AmB has also been associated with reduced maturation of viral glycoproteins and altered integrity of viral particles in the past, please determine actual infectious virus titers in lung tissue samples.

We fully agree with your perspective. Following the previously established modeling approach, we collected lung tissue and determined the influenza virus titer using the TCID₅₀ method. The results showed that the live influenza virus titer in the lung tissue of the AmB-treated group was significantly higher than that of the control group. This finding is consistent with the previous RT-qPCR results, supporting the conclusion that AmB promotes viral infection under *in vivo* conditions. We have now replaced the original **Fig. 1C** in the manuscript with the new live virus titer data. Due to the requirement of conducting SARS-CoV-2 live virus titer assays in a P3 laboratory, which is beyond our current experimental capabilities, we are unable to supplement this part of the experiment. We appreciate your understanding.

- 2) Presumably, the hospitalized patients that received AmB had more severe fungal co-infections compared to those that did not receive AmB. Please clarify as much as the records allow severity of the fungal coinfections in the different patient groups. If there is a positive correlation between administration of AmB and severity of fungal infection, poorer clinical outcome of this group must be anticipated independent of potential exacerbated viral

pathogenesis by AmB. Please outline your strategy for separation of AmB vs fungal co-infection-induced differences in outcome.

We thank the reviewer for pointing out the limitations of the initial clinical analysis. We fully acknowledge this issue and have accordingly made significant revisions to the clinical data analysis. The original analysis, based on 180 patients with influenza or COVID-19 complicated by Aspergillus infection, was limited by sample size, the number of patients exposed to AmB, and the control of treatment indications. These constraints made it difficult to adequately assess the independent association between AmB exposure and subsequent virus-related adverse outcomes. Therefore, to reduce bias, we have recollected data from all inpatients at China-Japan Friendship Hospital over a period of nearly a decade who had culture-confirmed Aspergillus infection and received systemic antifungal treatment (n = 1072) (**Fig. S3**). To address concerns regarding excessive deviation in the proportion of medication usage in certain years, **Table S1** shows annual counts and the proportion of patients treated with AmB among those with culture-confirmed Aspergillus infections requiring antifungal therapy.

Table S2 displays the baseline differences of the new cohort before propensity score matching. Consistent with your comments, the clinical data indicate that the condition of patients in the AmB group was indeed more severe. Therefore, we performed propensity score matching based on demographic characteristics, vital signs, underlying diseases, treatments, and clinical outcomes, and conducted multivariate logistic regression analysis on the matched cohort. The results showed that in the matched cohort, the incidence of viral infection after antifungal treatment was significantly higher in the AmB group than in the non-AmB group (21.55% vs. 7.76%, $p = 0.003$) (**Table S3**). In the multivariate model, which included age, sex, comorbidities, and immunosuppression-related medication use, AmB administration remained an independent risk factor (OR = 3.45, 95% CI 2.20 - 5.41, $p < 0.001$) (**Table S4**).

- 3) Very little particle binding (Fig. 2J) and very few internalization events (Fig. 2L) can be seen. It is unclear how these images correlate to the 5-10% positive cells claimed in the quantitations. Can you show fields of view that are more supportive of the quantitative data provided?

Your point is very important. To enhance the persuasiveness of our findings, we have completed the following work and updated the data:

We repeated the viral binding and internalization experiments and captured high-magnification images, which have now replaced the original **Fig.2J–M** in the manuscript.

In the above experiments, we included cycloheximide treatment to block host protein synthesis, ensuring that the detected NP signal originates from the invading virus rather than newly synthesized proteins. Additionally, necessary negative controls and neuraminidase controls were incorporated.

- 4) What are we looking at in fig. 5C? A meaningful legend explaining the color-intensity coding would be helpful.

Thank you for your kind reminder. We have revised **Fig. 5C** and provided detailed explanations in the figure legend. A brief summary of the figure's content is as follows:

1. Rows: Represent specific ceramide subspecies.
2. Columns: Represent different treatment groups (IAV vs. IAV+AmB), with four biological replicates per group.
3. Data Processing: Z-score normalization was applied to the values of each subspecies to eliminate baseline abundance differences across subspecies and highlight relative changes between treatment groups.
4. Color Coding: Red indicates a positive Z-score (above the overall sample mean), while blue indicates a negative Z-score (below the mean). The color intensity reflects the degree of deviation (Z-scores range approximately from -2 to +2).
5. Conclusion: After AmB treatment, the relative abundance of multiple ceramide subspecies in late endosomes generally increased, consistent with the upregulation of GCase activity.

5) Lines 74-75: the same antiviral mechanism of AmB has been proposed for HIV (i.e. PMID: 7983757). Worthwhile to add here.

Thank you. As noted in **lines 87–88** of the revised manuscript, we have incorporated the suggested citation into the **Introduction** as follows: “Intriguingly, this cholesterol-binding capacity enables AmB to inhibit cholesterol-dependent caveolin-mediated endocytosis, thereby suppressing viral entry of enterovirus 71, Japanese encephalitis virus, and even human immunodeficiency virus type 1 by blocking virus – cell fusion after CD4 binding (PMID: 7983757).”

6) Histopathology slides (Fig 1E and K) appear to be overstained.

We have adjusted the images in **Fig 1E and K** to RGB mode in the revised version, as the previous CMYK format may have contributed to the perception of overstaining in your initial review.

Reply to Reviewer #2:

The study by He et al. investigates the effects of amphotericin B (AmB), a widely used antifungal drug, on respiratory viral infections, specifically influenza A virus (IAV) and SARS-CoV-2. While AmB plays a critical role in treating invasive fungal infections, particularly in patients co-infected with viruses, the authors present evidence that AmB exacerbates viral infections by enhancing viral entry. Mechanistically, the study reveals that AmB binds to and activates glucocerebrosidase (GCCase), a lysosomal enzyme. This activation increases ceramide levels in late endosomes, promoting membrane fusion between the virus and the host cell, thereby facilitating viral genome release into the cytosol and enhancing replication. Notably, this effect is specific to late-penetrating viruses like IAV and SARS-CoV-2, and does not extend to early-penetrating viruses such as VSV, which fuse at early endosomes. The authors demonstrate that AmB enhances the entry of IAV and SARS-CoV-2 pseudoviruses across various cell types, independently of the antiviral host protein IFITM3. This enhancement depends on GCCase activity, as both pharmacological inhibition and genetic knockout of GBA1 suppress AmB-mediated viral entry. In vivo studies further support these findings: AmB treatment in IAV-infected mice and SARS-CoV-2-infected hamsters led to greater weight loss, higher viral loads, and more severe lung pathology compared to untreated controls. Additionally, retrospective clinical data from patients co-infected with influenza/SARS-CoV-2 and *Aspergillus* showed that those receiving systemic AmB had significantly longer ICU stays and mechanical ventilation durations, although mortality rates were not significantly different. At the molecular level, proteomic analysis revealed that AmB treatment upregulates RAB7 in a GCCase-dependent manner, contributing to enhanced viral fusion and entry. The authors propose a model in which AmB-induced GCCase activation and ceramide accumulation remodel late endosomal membranes to favor viral fusion.

These findings challenge current assumptions regarding the safety of AmB in virally co-infected patients and suggest that antifungal treatment regimens may need to be reassessed during viral outbreaks. Nevertheless, several major and a few minor issues remain to be addressed in this study.

Thank you for your recognition of our work and for your thorough review. Based on your suggestions, we have comprehensively repeated the live viral entry experiments, consistently incorporating cycloheximide to inhibit host translation. This ensures that the detected signals primarily originate from invading viral particles rather than newly synthesized progeny viruses. Additionally, as per your advice, we have included controls such as bafilomycin A1, NH₄Cl, and importazole, among others.

Major:

- 1) Although AmB treatment did not affect viral binding to the plasma membrane, as shown in Fig. 2I, an increase in NP levels was observed in AmB-treated cells at 2, 6, and 12 hpi (Fig. 2G). Notably, treatment with 1 µg/ml AmB significantly upregulated NP expression at 6 hpi, comparable to the effect seen with 0.3 µg/ml. However, at 2 hpi, 1 µg/ml AmB appeared to have no effect, and the increase at 12 hpi was modest. This discrepancy in the temporal response should be addressed by the authors. Additionally, the statistical significance of the 2 hpi data in Fig. 2G should be shown. The findings would be further strengthened by including known inhibitors of IAV entry, such as bafilomycin A1 or ammonium chloride, as positive

controls.

Thank you for your suggestion. We repeated the experiment using freshly prepared AmB, and the results are shown in the **revised Fig. 2G**. Bafilomycin A1 and NH₄Cl were also included as positive controls for endosomal acidification/invasion inhibition. The results indicate that 1 µg/mL AmB significantly increased intracellular NP levels at both 6 hpi and 12 hpi. While 0.3 µg/mL AmB did not show a significant difference at 6 hpi, a significant difference was observed at 12 hpi. Therefore, we believe that the effect of 1 µg/mL is more robust and significant. The previous inconsistencies in effects across different time points may have been due to variations in drug freeze-thaw cycles or preparation methods. With the use of freshly prepared drugs and positive controls, the conclusions are now more robust.

- 2) Since IAV entry in A549 cells occurs rapidly, with most viral particles reaching the late endosomes within an hour, earlier time points should be examined. Assessing later time points such as 6 h or 12 h may not accurately reflect entry-related events.

We fully agree with the reviewer's suggestion regarding focusing on the early time window and have conducted a more detailed temporal analysis. The purpose of **Fig. 2G** was to macroscopically evaluate the effective stage of AmB during the entire infection process, hence the selection of three time points: 2, 6, and 12 hpi. To specifically address the impact of AmB on entry-related events, we have additionally performed a more refined co-localization temporal experiment (**revised Fig. 3**). The objectives of these two experiments differ: Fig. 2G is used to determine the overall effective time window, while Fig. 3 is designed to analyze the detailed kinetics of endosomal transport, membrane fusion, and nuclear entry.

- 3) In viral entry assays, cycloheximide is commonly used to block host translation, ensuring that detected NP (or other viral components) originates from incoming virions rather than newly synthesized proteins. However, it appears that cycloheximide was not used in the current experiments. To accurately assess viral entry, all relevant assays should be repeated with cycloheximide treatment.

Your suggestion is really helpful. We have now incorporated cycloheximide treatment into all live viral entry-related experiments, including **Fig.2 I–M**, **Fig.3 A–J and L**, **Fig.5E**, **Fig.S6A**, and **Fig.S10A**. The revised manuscript has been updated accordingly with the new results obtained under cycloheximide treatment. Overall, the addition of cycloheximide does not alter our original conclusions: even under cycloheximide treatment, AmB still significantly promotes membrane fusion of the virus with late endosomes, leading to faster viral release from late endosomes and subsequent nuclear entry.

- 4) The images in Fig. 2J and 2K do not convincingly demonstrate viral binding. For a proper IAV binding assay, surface-bound viruses should be visualized at higher magnification under unpermeabilized conditions using a membrane or cell surface marker. The NP signals appear discrete and patchy, which is unexpected at an MOI of 0.1, where viral particles are typically distributed across most cells, not limited to a few “NP-positive” cells. Quantification should

be based on relative fluorescence intensity of viral proteins at the plasma membrane, ideally compared to controls such as no-virus or neuraminidase-treated cells.

As per your suggestion, we redesigned and performed the binding experiment. Specifically, under non-permeabilized conditions, sialic acid was used as a cell surface marker, and imaging was conducted under high-magnification confocal microscopy. Negative controls, including no-virus and neuraminidase pretreatment groups, were also established. Quantitative analysis of membrane-associated NP was performed using ImageJ. The results indicate that AmB treatment does not affect the binding of IAV to the cell surface, whereas neuraminidase significantly reduces membrane binding (**revised Fig.2J & K**).

- 5) Influenza A virus (IAV) is known to enter cells via both clathrin-mediated and clathrin-independent endocytic pathways. However, the effect of AmB has been assessed only in the context of clathrin-mediated uptake, using transferrin as a marker. This does not account for other clathrin-independent mechanisms such as macropinocytosis, which IAV is also known to utilize. The authors should evaluate the impact of AmB on additional endocytic pathways, particularly macropinocytosis. Moreover, several experiments lack appropriate controls, which are essential for drawing robust conclusions.

We thank you for this important and insightful suggestion. Based on this, we have expanded our investigation to assess other endocytic pathways. We evaluated the pathways using Alexa Fluor 488-conjugated transferrin (clathrin-mediated endocytosis), TMR-dextran 70 kDa (macropinocytosis), and Alexa Fluor 555-conjugated cholera toxin subunit B (caveolin-mediated endocytosis), with Pitstop 2, EIPA, and M β CD serving as corresponding inhibitor positive controls. Flow cytometry results showed that the respective inhibitors significantly reduced endocytic efficiency, while AmB did not significantly affect clathrin-mediated endocytosis, macropinocytosis, or caveolin-mediated endocytosis, consistent with our original conclusions (**revised Fig. S9**).

- 6) The manuscript frequently reports colocalization using Manders' coefficient; however, it does not specify whether M1 or M2 values are being used. This distinction is critical for proper interpretation of the data and should be clearly indicated in all relevant instances throughout the manuscript.

Thank you for pointing this out. To measure the proportion of viral signal present in specific cellular structures (such as the cell membrane, early/late endosomes, or the nucleus), we utilized Manders' M2 coefficient across all relevant analyses. In this calculation, channel A was designated for the viral signal, and channel B for the structural marker. We have now included this specification in the **Methods** section and in the **figure legends for Fig. 2 and Fig. 3**.

- 7) The authors claim that at 3 hpi, IAV escapes more rapidly from Rab7-positive late endosomes in AmB-treated cells compared to controls. While Fig. 3B shows increased NP signal in Rab7-negative regions of AmB-treated cells, it is highly unlikely that incoming viruses at this time point would still remain in the cytosol without having entered the nucleus. Based on current understanding, vRNPs released after fusion typically undergo rapid nuclear import, rather than

persisting in the cytosol. Moreover, at the current image resolution, it is difficult to accurately assess NP distribution, and the vRNPs appear largely punctate. To better understand whether the vRNPs are still within early endosomes, co-staining of NP with early endosomal markers such as EEA1 is recommended. Additionally, higher-resolution images are essential to reliably determine endosomal localization or escape of viral particles, especially under cycloheximide treatment.

Regarding the issue of NP and endosomal localization, we have consolidated our responses to your points 7 and 8 as detailed below.

- 8) To substantiate the claim that AmB treatment accelerates viral escape from late endosomes, a time-course analysis with finer temporal resolution, rather than relying solely on 2 and 3 hpi, should be conducted. Additionally, more rigorous quantitative analysis is necessary, along with appropriate positive controls such as bafilomycinA1 or ammonium chloride. As mentioned above, high-resolution images that clearly depict endosomal structures and viral localization are essential to support this conclusion.

As suggested, we performed co-localization staining of NP with EEA1 or RAB7 under cycloheximide treatment at 1, 2, and 3 hpi. The results showed that at 1 hpi, NP-EEA1 co-localization was predominant, indicating that the virus primarily localized to early endosomes. At 2 hpi, NP-RAB7 co-localization emerged, reflecting a shift to late endosomal localization. By 3 hpi, the IAV group still exhibited a relatively high level of NP-RAB7 co-localization, whereas the IAV+AmB group showed a significant reduction in NP-RAB7 co-localization and a tendency for NP to migrate toward the nuclear region. Additionally, we used bafilomycin A1 as a positive control for inhibiting endosomal fusion. In this group, NP remained trapped in late endosomes, and nuclear entry was blocked. These higher-resolution and temporally refined data demonstrate that AmB promotes faster viral escape from late endosomes and facilitates nuclear entry by 3 hpi, supporting our original conclusion. The above results are now presented in **revised Fig. 3A - F**.

- 9) In Fig. 3E, since IAV fusion typically occurs between 1 and 2 hpi, the absence of a fusion signal at 2 hpi in control samples requires clarification. Likewise, the near-complete absence of SP-DiOC18 signal at 4 hpi in control cells, as shown in Fig. 3F, is highly unlikely and needs explanation. It is improbable that no SP-DiOC18-positive particles would be detected in untreated cells. Furthermore, the SP-DiOC18 signal in the images provided appears diffusely distributed throughout the cytosol, rather than localized to endosomal compartments as expected. These observations raise serious concerns about the reliability of this otherwise well-established IAV fusion assay. The experiment should be repeated under cycloheximide treatment, include appropriate positive controls, and be supported by high-magnification images that clearly resolve endosomal localization.

In response to the reviewer's inquiries regarding experimental details, we have refined the methodology and repeated the fusion assay. Under cycloheximide treatment, we reperformed the dual-labeled R18/SP-DiOC18 viral infection experiment, including bafilomycin A1 as a positive control for fusion inhibition. We also optimized the SP-DiOC18 labeling and washing procedures

to enhance imaging resolution and reduce nonspecific diffuse signals. The results show minimal fusion events at 1.5 hpi, while at 3 hpi, the IAV+AmB group exhibited a significant increase in fusion signals, which was markedly suppressed in the bafilomycin A1-treated group. These findings align with our original conclusion that AmB promotes viral fusion with late endosomal membranes. The updated results are presented in the **revised Fig. 3G and H**.

Moreover, variations in the reported time windows for fusion across studies may stem from differences in viral strains, cell lines, and experimental conditions. Therefore, the limited fusion observed at 1.5 hpi does not necessarily contradict the literature. For instance, in the following two references, substantial membrane fusion was also observed only at 3 hpi.

Sci China Life Sci. 2024 Mar;67(3):579-595. doi: 10.1007/s11427-023-2471-4. (PMID: 38038885)
Fig. 4A & B

mBio. 2015 Jun 9;6(3):e00297. doi: 10.1128/mBio.00297-15. (PMID: 26060270)
Fig.4A & B

redacted

- 10) The authors state (line 202) that AmB enhances nuclear accumulation of viral NP. However, Fig. 3G shows a strong NP signal in the cytoplasm at 6 hpi. At this time point, such signals are typically attributed to newly synthesized NP, which localizes to both the nucleus and cytoplasm. Therefore, without cycloheximide treatment to inhibit viral protein synthesis, it is not possible to specifically assess vRNP nuclear import during the early entry phase. Additionally, evaluating vRNP nuclear import at late time points such as 9 or 12 hpi is unnecessary and potentially misleading. Instead, a time-course analysis focusing on earlier time points, up to 4 or 5 hpi, would be more appropriate, as nuclear import of vRNPs is known to occur within this window. Appropriate controls such as importazole (importin beta inhibitor) or bafilomycin A1 should be included.

Thank you for your suggestion. We assessed viral nuclear entry at 5 hpi under cycloheximide treatment, while using importazole, bafilomycin A1, and NH₄Cl as positive controls to inhibit viral nuclear entry. The results indicate that the nuclear NP level in the IAV+AmB group was higher than that in the IAV group. Additionally, importazole, bafilomycin A1, and NH₄Cl significantly suppressed nuclear viral signals, confirming that AmB indeed promotes viral nuclear entry at this time point rather than inhibiting viral replication. These updated results are now presented in the **revised Fig. 3I, J, and L**.

- 11) To support the claim that AmB promotes IAV escape from late endosomes by enhancing fusion at the endosomal membrane, an acid bypass infection assay should be performed. This approach induces low pH-mediated viral fusion at the plasma membrane, allowing viral genomes to reach the nucleus without endosomal trafficking. If AmB acts specifically at the level of late endosomes, NP expression levels following plasma membrane fusion should be comparable between AmB-treated and control cells.

As suggested, we performed the acid-bypass experiment. The results indicate that under conventional infection conditions, the NP level in the AmB group was significantly higher than that in the control group. However, under acid-bypass conditions, no significant difference in NP levels was observed between the AmB and control groups. Therefore, we conclude that when viral entry

does not depend on the late endosomal pathway, AmB no longer exerts an effect. This finding is consistent with the results showing that AmB does not affect viral entry from early endosomes (**Fig. 4C**), confirming that the infection-promoting effect of AmB specifically occurs during the stage of viral fusion with late endosomal membranes. These updated results are now presented in the **revised Fig. S10A**.

- 12) A time-of-addition assay for AmB should be performed, particularly during early time points and in the presence of cycloheximide, to determine whether the observed enhancement in NP expression is truly due to accelerated viral escape from late endosomes.

As suggested, we performed a time-of-addition experiment by administering the drug at -1 h, 0 h, +1 h, +2 h, and +4 h relative to infection, with samples collected at 6 hpi. All experiments were conducted under cycloheximide treatment. The results showed that NP levels in the groups treated with AmB at -1 h, 0 h, and +1 h were significantly higher than those in the control group, whereas no significant differences were observed in the groups treated at +2 h and +4 h compared to the control. This indicates that AmB acts during the early stages of infection. These results are now presented in **revised Fig. S6A**. However, the time-of-addition experiment alone is insufficient to demonstrate that AmB promotes late endosomal escape of the virus. Therefore, we have integrated the findings from the time-of-addition, co-localization, membrane fusion, and acid-bypass experiments to collectively support the conclusion that AmB facilitates viral escape from late endosomes.

- 13) In Fig. S7A, LysoSensor is used to assess endosomal pH; however, critical controls such as bafilomycin A1 or ammonium chloride, which are known to raise endosomal pH, are missing. Additionally, the baseline pH of untreated cells is not indicated and should be included to allow proper interpretation of the results.

As suggested, we have added untreated and bafilomycin A1-treated groups as controls in this experiment. The results indicate that bafilomycin A1 significantly increases endosomal pH, which aligns with established knowledge. In contrast, AmB treatment causes a slight elevation in endosomal pH, and this modest effect is consistent with our original conclusion. This suggests that the fusion-promoting effect of AmB is not mediated through a reduction in endosomal pH. These updated results are now presented in the **revised Fig. S10B and C**.

- 14) In Fig. 5A, AmB treatment is applied for 4 h to measure GCCase activity, while in other experiments it is applied for only 1 h prior to IAV infection. Measuring GCCase activity after 1 h of AmB treatment and then at intervals up to 4–6 h would provide a better understanding of the kinetics.

In the original manuscript, we measured GCCase activity at 4 hours because AmB significantly promotes viral release from late endosomes and membrane fusion at 3 hpi. Since AmB was pre-treated for 1 hour prior to infection and remained in the culture medium afterward, we considered the total cellular drug exposure time to be 4 hours, when its effects became evident. To further clarify this, we supplemented the study by measuring GCCase activity in A549 cells treated with AmB at

different time points: 0, 15, 30, 60, 120, and 240 minutes, followed by curve fitting. The results indicate that GCCase activity increased rapidly after AmB treatment, peaking at approximately 1 hour and then reaching a plateau, with high activity still maintained at 4 hours. These findings are now presented in **revised Fig. S12A**.

15) The authors convincingly show that GCCase activity in purified endosomes is enhanced upon AmB treatment and propose that this increase promotes viral escape from late endosomes via elevated ceramide levels. While the effect of AmB on GCCase activity appears robust, the current data do not definitively establish whether this impact occurs specifically during viral entry. In the absence of reliable viral entry assays, it remains possible that elevated GCCase activity in other cellular compartments contributes to enhanced viral gene expression at post-entry stages. To address this, experiments should be carefully designed to distinguish whether increased ceramide production directly facilitates viral entry or instead influences downstream steps in the viral life cycle.

To distinguish whether GCCase activity affects viral infection during the entry or post-entry stages, we conducted the following additional experiments:

(1) Infection assay in GBA1 KO cells: Live viral infection was performed using WT and GBA1 KO 293T cells, and viral load was measured by RT-qPCR. The results showed that GBA1 knockout significantly suppressed viral infection (**revised Fig. 5D**).

(2) Time-of-addition experiment using the GCCase inhibitor CBE: CBE was added at -1 h, 0 h, +1 h, +2 h, and +4 h relative to infection, and samples were collected at 6 hpi. The results indicated that early administration of CBE at -1 h and 0 h significantly reduced NP levels, whereas no significant effect was observed when CBE was added at +1 h or later time points (**revised Fig. 5E**).

(3) Late-stage CBE administration assay: CBE was administered at 6 hpi, and viral load was measured at 24 hpi. The results showed that late administration of CBE had no significant effect on viral load (**revised Fig. 5F**).

Additionally, combining these findings with the pseudovirus experiments in **Fig. 5G–I**, which specifically investigate the viral entry stage, the data collectively suggest that GCCase activity primarily influences the early stages of viral infection, with no significant effect on subsequent phases of the viral life cycle.

16) In addition to testing pseudoviruses in GBA1 KO or KD cells, WT viruses such as PR8 should be tested.

As suggested, we performed live PR8 viral infection experiments using WT and GBA1 KO 293T cells. The results showed that GBA1 knockout significantly suppressed viral infection and attenuated the infection-promoting effect of AmB (**revised Fig. 5D**).

17) In addition to Rab7 and LAMP1, the PCA analysis in Fig. 5 should incorporate other lysosomal

hydrolases implicated in late endosomal function and viral entry. Including these markers would provide stronger support for the claim of lysosomal remodeling.

This is a highly effective suggestion. In the supplementary materials, we have expanded and presented more endosomal/lysosomal-related proteins quantified through data-independent acquisition-based quantitative proteomics (**revised Fig. S13**). In addition to RAB7, LAMP1, and NPC1 shown in Fig. 6, Fig. S13 includes CAV1, CAV2, MTMR9, KXD1, COMMD2, MOSPD2, OSBPL11, AP5S1, RILP, and GABARAP. Overall, AmB indeed significantly alters the composition of endosomal proteins, supporting our conclusion regarding endosomal remodeling.

18) In Fig. 6J, quantification of endosomal marker levels from immunoblots is presented without showing the corresponding loading control. This is essential for accurate interpretation.

Protein concentrations of endosome samples were first measured using the BCA assay to ensure equal protein loading for Western blot analysis. To further address the reviewer's concern regarding loading controls, we performed silver staining, which confirmed comparable total protein levels across all samples (as shown in the picture below). Thus, the normalization based on BCA measurements and validated by silver staining supports the reliability of the Western blot quantitation. This information has been added to the **Methods** section and the **figure legend**.

19) As already mentioned, appropriate controls are missing in many of the experiments.

We have sequentially supplemented the key controls you indicated, such as bafilomycin A1, NH₄Cl, neuraminidase, importazole, Pitstop2, EIPA, and M β CD, among others. Overall, the inclusion of these additional controls did not alter the original conclusions but rather enhanced their rigor.

Minor:

20) In line 169, the figure reference should be corrected from Fig. 2G to Fig. 2J.

Thank you for pointing this out. We have corrected the erroneous figure citations in the text and

thoroughly verified all figure and table references in the revised manuscript to ensure consistency.

21) The sequence of subpanels in Fig. 5 should be reorganized for logical flow.

We have adjusted the panel order in **revised Fig. 5** according to your suggestion to better align with the logical flow, and have correspondingly reordered the **figure legend** and revised the accompanying text.

Reply to Reviewer #3:

The authors start by setting up an influenza A mice model and SARS-CoV-2 hamster models which are pre-treated with IV amphotericin B one hour prior to, and then at 24 hr and 72 hr after virus challenge. Compared with controls without amphotericin B treatment, significantly higher weight loss, lung viral load or viral antigen positive cells, histopathological changes of necrotizing pneumonia and serum creatinine levels due to subclinical renal damage were found at day 2 and 5 post-infection. With this finding, they conducted a retrospective clinical study using 2017 to 2019 influenza A patients, and 2022 to 2023 SARS-CoV patients with fungal test positive. 17 patients with Amphotericin B treatment were found to have significantly higher rate of ICU admissions, ICU stay, duration of mechanical ventilation (but not the mortality rate at 28 day) when compared to 163 without amphotericin B treatment. Then they showed that A549 cell line pretreated with Amphotericin B produced a higher viral load and NP expression after infection by PR8 H1N1 starting at 6 hr postinfection. Cell entry of H5 and SARS-CoV-2 pseudovirus was also augmented in another 3 cell lines by Amphotericin B and Nystatin independent of IFITM3 expression as demonstrated by knockdown and overexpression experiments. Flow cytometry and immunofluorescence analysis showed that Amphotericin B enhances viral entry independent of binding, endocytosis or IFITM3 restriction. Time-resolved co-localization analyses at 3 hr postinfection showed that influenza virions in Amphotericin B treated cells had significantly reduced retention within late endosomes and showed accelerated dispersal toward the nuclear region which suggested an increased escape from late endosomes. A dual fluorescent PR8 virus fusion assay confirmed that Amphotericin B promotes PR8 virus-endosome fusion with concomitant alkalinisation of endosomes. Furthermore, amphotericin pretreatment significantly enhanced the entry of HIV/SARS CoV 2 and HIV/MERS CoV pseudoviruses as shown by luciferase assays. Finally, they showed that Amphotericin B-biotin conjugate specifically pull down glucocerebrosidase (encoded by GBA1). Moreover, Amphotericin B treatment significantly enhanced glucocerebrosidase activity in A549 and other cell lines. Chemical inhibition of glucocerebrosidase by conduritol epoxide and CRISPR-Cas9 knockout of GBA1 in 293T cell line reduced enzymatic activity and the amphotericin B mediated enhancement of virus entry in pseudovirus assays of influenza and SARS-CoV-2. Proteomic analysis showed that amphotericin B upregulated late endosomal marker RAB7, caveolin-1 and RAB interacting lysosomal protein on proteomic analysis and Western blot analysis, whereas GBA1 gene deletion downregulated all these markers. The authors concluded that amphotericin B binding triggers a conformational activation of glucocerebrosidase which leads to ceramide accumulation and upregulation of RAB7 and therefore remodels endosomal architecture. These changes promote the fusion of the viral envelope with late endosomal membranes. Therefore, glucocerebrosidase-regulated ceramide synthesis and endosomal maturation can be novel pharmacological targets for modulation of virus infection.

Strength

1. The clinical question is important though mould infection after influenza and SARS-CoV-2 are relatively uncommon unless there are major underlying lung disease, prolonged invasive ventilation and immunosuppression.
2. The experiments are well done, and the paper is well written.

Thank you for your thorough review and constructive feedback.

To enhance the clinical applicability of our study, while retaining the pretreatment model for mechanistic analysis, we have supplemented the research with both *in vitro* and *in vivo* experiments involving post-infection drug administration. Additionally, we have conducted extensive re-statistics and rigorous statistical analyses on the clinical cohort. Our study primarily serves as a reference for antifungal medication in patients with fungal infections. The findings indicate that patients treated with AmB face a higher risk of subsequent respiratory viral infections.

- 1) In the usual clinical situation, the patient usually has a long period of influenza A or SARS-CoV-2 disease before the onset of Aspergillus or Mucorales infection, and a significant proportion has been given steroid or biologics immunosuppression before the onset of these mould infections. It would be wrong to give amphotericin B pre-treatment before virus challenge to cell lines and animals.

Thank you for highlighting this important clinical pattern. From a clinical epidemiological perspective, since the number of viral infections far exceeds that of fungal infections, most secondary infection cases indeed begin with a viral infection, followed by secondary infections such as Aspergillus. However, among hospitalized patients with fungal infections, there are also cases in which patients are exposed to respiratory viruses and develop secondary infections while receiving antifungal treatment. For instance, as noted in the newly added **reference 9** in the **Introduction**, "patients with established invasive pulmonary aspergillosis (IPA) who subsequently acquire SARS-CoV-2 exhibit markedly worse outcomes" (Ann Clin Microbiol Antimicrob. 2025 Jun 18;24(1):38. doi: 10.1186/s12941-025-00805-8. PMID: 40533753).

In the revised manuscript, we have recollected data from all culture-confirmed Aspergillus-positive inpatients who received systemic antifungal treatment at China-Japan Friendship Hospital over the past 10 years (n = 1072). The results show that 112 patients developed viral infections after initiating antifungal therapy, indicating that post-antifungal viral infections do indeed occur in real-world settings. These findings have been incorporated into the **Results** section.

Therefore, this study has two primary objectives. First, to investigate the mechanism by which AmB promotes viral infection, and second, to explore its clinical relevance by assessing the association between AmB use and secondary viral infection outcomes in real-world settings. For mechanistic validation, we employed a "drug pretreatment" model to ensure sufficient drug delivery to endosomes and its effect prior to viral entry—a commonly used experimental paradigm that most directly elucidates the impact of drugs on various stages of cellular entry. For clinical relevance analysis, we expanded and re-evaluated a clinical cohort (details provided below).

- 2) In the retrospective case control study of the clinical patients treated or not treated with amphotericin B, the authors have not provided sufficient clinical data in terms of their infection by which mould species and whether the documentation by culture, antigen or PCR test. They have not provided the data in terms of time of onset of mould infection and the virus infection, and most importantly the amount and duration of immunosuppression by steroid and biologics.

Thank you for highlighting this important need for information. To improve the interpretability and reproducibility of the clinical analysis, we have made significant revisions to the clinical section:

(1) Cohort Expansion and Inclusion Criteria: We recollected and included all culture-confirmed, Aspergillus-positive inpatients who received systemic antifungal therapy at China-Japan Friendship Hospital over nearly a decade (n = 1072). All cases were confirmed based on respiratory sample cultures (**new Fig. S3**).

(2) Immunosuppression Data: We systematically collected information on the use of glucocorticoids and biologics, and incorporated these variables into both the propensity score matching and multivariate regression models to control for this critical confounder to the extent possible (**new Table S2–4**).

(3) Temporal Information: We documented the start time of antifungal therapy and the subsequent onset time of any viral infection. This information was used to define the outcome variable "viral infection after antifungal therapy," which served as the primary outcome.

(4) Main Findings: In the expanded and propensity score-matched cohort, the incidence of viral infection after antifungal therapy was 21.55% in the AmB group versus 7.76% in the Non-AmB group (p = 0.003) (**new Table S3**). In a multivariate logistic regression model adjusted for covariates including age, sex, major comorbidities, and the use of glucocorticoids/biologics, AmB use remained an independent risk factor (OR = 3.45, 95% CI 2.20 – 5.41, p < 0.001) (**new Table S4**).

These expanded data do not refute the clinical reality that "most cases involve viral infection occurring first." However, they indicate that in real-world hospital settings, there is a risk of "viral infection after antifungal therapy." Furthermore, even after statistically controlling for multiple confounding factors, AmB use remains associated with a higher incidence of secondary viral infection.

3) Therefore, all the virus infection of cell lines and animal experiments must also be repeated and findings validated by giving amphotericin B after virus challenge. For example, the amphotericin B should be given at least 48 hours after virus challenge which is the usual or normal clinical scenario.

Thank you for this relevant suggestion. We have also considered the alignment between the experimental model and clinically relevant time windows, and accordingly supplemented the following experiments with the results outlined below:

(1) In vitro post-infection treatment assay: We performed additional experiments in A549 cells where AmB was administered at 24 h and 48 h post-infection, with viral load measured 24 hours after drug treatment. The results showed no significant effect on viral load in the post-treatment (24–48 h) group (**revised Fig. S6B**).

(2) In vivo post-infection treatment assay: In the IAV-infected mice, AmB was administered at 48 h, 72h, and 96 h post-infection, and samples were collected on day 5 post-infection for analysis. The results indicated no significant differences between the post-AmB treatment group and the IAV-infected control group in terms of body weight, lung viral load, serum biochemistry, or pathological injury. Thus, no infection-promoting effect of AmB was observed under these late-stage treatment conditions, consistent with the in vitro data (**revised Fig. S6C–H**).

These findings demonstrate that the infection-promoting effect of AmB is highly time-dependent. When administered during the late stages of viral infection, AmB does not significantly enhance viral infection. This observation is highly consistent with our proposed mechanistic model, wherein AmB promotes viral fusion by activating GCase, increasing endosomal ceramide levels, and remodeling late endosomal membrane proteins.

RESPONSE TO REVIEWERS' COMMENTS (NCOMMS-25-38479A)

We sincerely thank the reviewers for their time and effort in assessing our revised manuscript. We are delighted to learn that our responses and additional data have fully addressed your concerns. We deeply appreciate your constructive comments throughout the review process, which have significantly improved the quality and clarity of our work.

Reply to Reviewer #1:

The authors have made a major effort in their response to previous critiques, adding a substantial amount of new experimental data. My concerns have been addressed.

We sincerely thank the reviewer for the positive evaluation of our revised manuscript. We are glad that the additional experimental data have successfully resolved your concerns. We express our deep gratitude for your time and for recognizing our efforts to improve the study.

Reply to Reviewer #2:

I have carefully evaluated the revised manuscript. The authors have now adequately addressed all the points raised by me in my previous review. The revisions have led to a significant improvement in the overall quality of the work. I have no further major concerns and believe the manuscript is now suitable for publication.

We are grateful to the reviewer for the careful re-evaluation of our work and for the encouraging comments. We truly appreciate your acknowledgement that our revisions have led to a significant improvement in the quality of the manuscript. Your constructive suggestions during the review process have been invaluable, and we thank you for recommending our work for publication.

Reply to Reviewer #3:

The authors have adequately addressed my concern with additional analysis of clinical data, and repeat cell line and animal experiments with reversal of the order of antifungal treatment. I have no additional comments.

We would like to thank the reviewer for the positive feedback. We are pleased to know that our additional clinical analysis and repeated experiments have adequately addressed your concerns. We appreciate your rigorous review and your support for the publication of this study.